# RETROSPECTIVE SPARSE ATTENTION FOR EFFICIENT LONG-CONTEXT GENERATION

**Seonghwan Choi**\*, **Beomseok Kang**\*, **Dongwon Jo, Jae-Joon Kim**
Seoul National University
`{csh3695,beomseok,dongwonjo,kimjaejoon}@snu.ac.kr`

## ABSTRACT

Large Language Models (LLMs) are increasingly deployed in long-context tasks such as reasoning, code generation, and multi-turn dialogue. However, inference over extended contexts is bottlenecked by the Key-Value (KV) cache, whose memory footprint grows linearly with sequence length and dominates latency at each decoding step. While recent KV cache compression methods identify and load important few tokens, they focus predominantly on input contexts and fail to address the cumulative attention errors that arise during long decoding. In this paper, we introduce RetroAttention, a novel KV cache update technique that retrospectively revises past attention outputs using newly arrived KV entries from subsequent decoding steps. By maintaining a lightweight output cache, RetroAttention enables past queries to be efficiently supplemented with more contexts, while incurring minimal latency overhead. This breaks the fixed-attention-output paradigm and allows continual correction of prior approximations. Extensive experiments on long-generation benchmarks show that RetroAttention consistently outperforms state-of-the-art (SOTA) KV compression methods, increasing effective KV exposure by up to 1.6× and accuracy by up to 21.9%. The code is available at: `https://github.com/csh3695/RetroAttention`

## 1 INTRODUCTION

With the growing demand for long-context tasks such as reasoning (Jaech et al., 2024; Guo et al., 2025; Comanici et al., 2025), code generation (Chen et al., 2021; Jiang et al., 2024), and multi-turn dialogue (Yi et al., 2024), Large Language Models (LLMs) are increasingly expected to handle extended sequences of input and output. However, supporting such long-context scenarios imposes substantial computational challenges. A key constraint is the Key-Value (KV) cache, whose memory footprint grows linearly with context length. In practice, the KV cache can occupy several GBs, and fetching it at every decoding step often becomes the main latency bottleneck for inference (Zadouri et al., 2025; Yuan et al., 2024). As a result, efficient KV cache management has emerged as a critical technique for enabling scalable long-context inference in modern LLMs.

Recent studies have demonstrated that only a small portion of the KV cache plays a critical role in attention computation (Ribar et al., 2023; Zhang et al., 2023; Li et al., 2024). Motivated by this, several KV cache compression techniques have been proposed to identify and only retain important tokens (Liu et al., 2023; Cai et al., 2024; Singhania et al., 2024; Chen et al., 2024b). However, these methods primarily focus on processing long-context *inputs*, rather than *outputs*. Long generation introduces a distinct challenge: errors from approximated attention, due to evicted KV entries, recursively accumulate in the model's hidden states over extended decoding steps. As illustrated in Figure 1(b), which evaluates generation quality on PG-19 (Rae et al., 2019), a long-range language modeling benchmark, the performance gap between models using full and compressed KV cache is initially marginal but becomes substantial as generation progresses. This trend highlights the inherent weakness of existing approaches, which emphasize token selection for the current decoding step while leaving previously decoded tokens unchanged (Yang et al., 2024; Tang et al., 2024; Xiao et al., 2024; Liu et al., 2024a). A naive solution would be to allocate more KV cache budget (*i.e.*, load more KV entries), but doing so increases memory usage and latency, defeating the main purpose

---

\*Both authors contributed equally to this work.

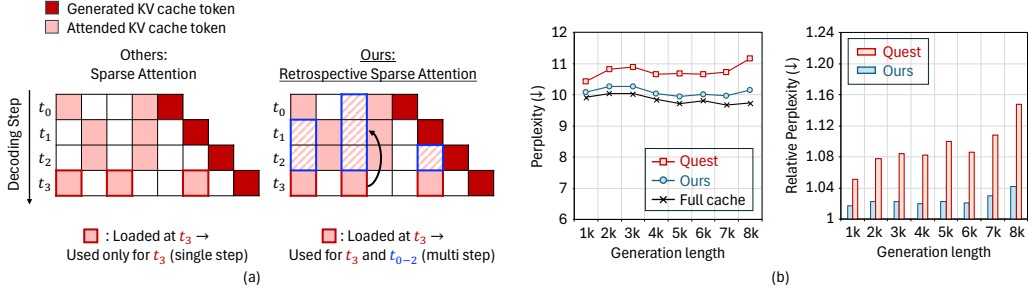

Figure 1: **High-level idea of RetroAttention**. (a) illustrates the key distinction between conventional sparse attention (*e.g.*, Quest) and the proposed Retrospective Sparse Attention. In our approach, KV entries that are currently retrieved ($t_3$) but were unseen in previous decoding steps ($t_0$-$t_2$) are reused to augment past attention computations. (b) demonstrates the average Perplexity depending on the range of generation lengths and relative Perplexity between full and compressed KV cache models (right), indicating significant quality degradation as generation continues.

of compression. This raises a fundamental research question: How can we mitigate the cumulative attention errors during long generation without increasing the KV cache budget?

In this paper, we propose a novel KV cache *update* technique, called **Retro**spective Sparse **Attention** (**RetroAttention**), which enhances approximated attention for past Queries by leveraging future KV entries arriving in subsequent decoding steps (see Figure 1(a)). RetroAttention builds on a query-aware non-eviction KV selection scheme, where the model assumes that some relevant KV entries may not be observable at the time a Query is processed but can be loaded later during decoding. To efficiently leverage this behavior, we allocate a lightweight *output cache* that stores the attention outputs of past Queries. As new KV entries arrive in subsequent steps, we retrospectively compute additional attention outputs for past Queries to update their previous outputs in the cache. This process effectively compensates for errors in the initial attention outputs, thereby improving the quality of KV representations passed to deeper layers. Consequently, each Query benefits from exposure to more KV entries than the static cache budget would allow, without extra KV memory traffic. Our key contributions are as follows:

- We propose **RetroAttention**, a novel KV cache update method for long-context generation that *retrospectively refines* past attention outputs using newly retrieved KV entries. RetroAttention leverages a lightweight output cache, enabling efficient access and updates of past computations as future KV entries arrive.

- We exploit the *memory-bound* nature of long-context inference, where the proposed retrospective updates incur negligible latency. We analytically demonstrate this property using Arithmetic Intensity (AI) and empirically confirm that RetroAttention adds only marginal end-to-end latency overhead.

- We extensively validate that retrospective updates effectively expand the number of KV entries exposed to queries (the *effective KV budget*) by up to $1.6\times$, without increasing the actual KV budget. RetroAttention achieves up to 21.9% and on average 5.6% accuracy gains over SOTA methods on long-generation benchmarks.

## 2 PROPOSED APPROACH

### 2.1 BACKGROUND

**Dynamic Sparse Attention.** RetroAttention adopts Quest's KV cache selection strategy (Tang et al., 2024) to address evolving importance of KV cache pages during decoding. Quest abstracts each KV cache page into $K_{\min}$ and $K_{\max}$, the element-wise minima and maxima of Key vectors within a page, which highlight extreme feature values. The importance of the $j$-th cache page to a given Query is then estimated as:

$$\text{score}_j(Q) = \sum_i \max(Q_i K_{\min,i}^j, \ Q_i K_{\max,i}^j) \tag{1}$$

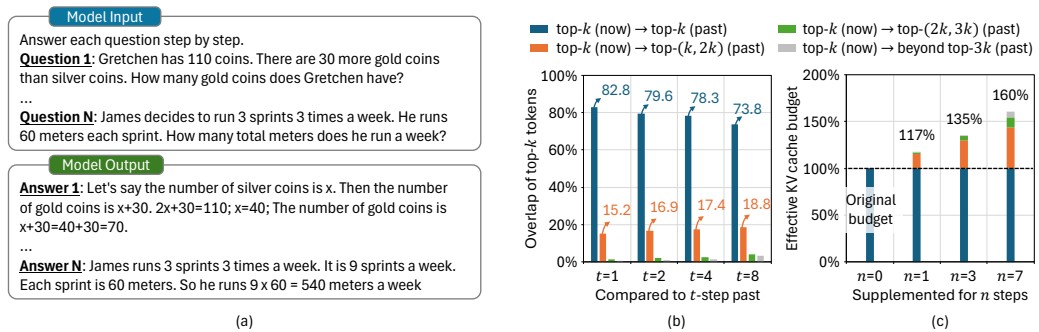

(a)          (b)          (c)

Figure 2: **Long-generation accuracy analysis**. (a) illustrates an example in LONGGENBENCH, a long-generation benchmark, where the input prompt includes concatenated questions. (b) presents the semantic correlation of the current top-$k$ KV entries in the $t$-step prior decoding, categorized into different top-$k$ intervals, demonstrating how often current important tokens would have been important in the past. (c) analyzes the effective KV cache budget (*i.e.*, total KV entries exposed to a given Query) when supplemented with entries from $n$ future decoding steps. Note that this growth may not scale linearly to $n$, as supplemented KV entries can be duplicated across updates. Results are experimented with LLAMA3.1-8B-INSTRUCT and GSM8K in LONGGENBENCH.

where $i$ indexes the feature dimensions. This can be computed without loading full pages and enables efficient top-$k$ selection of relevant pages at each decoding step.

This approach features that unselected KV entries are not permanently discarded—available for future retrieval if they become relevant. While Quest emphasizes the Query-dependent variability of optimal KV entries, we shift the focus to their *reusability*, using currently loaded entries to retrospectively refine past attention outputs.

## 2.2 MOTIVATIONAL STUDY

Our motivation primarily arises from long-generation scenarios, where even small amounts of KV cache supplementation can accumulate over many decoding steps and ultimately lead to non-trivial performance gains. We emphasize that the dependency on the KV cache budget differs substantially between long-context *input* tasks and long-context *output* tasks.

**Long-generation Tasks are More Sensitive to KV Budget.** Our observation shows that, in LONGBENCH (long-input, short-output), Quest with only a 5% KV budget already achieves accuracy (46.3%), comparable to the full-cache baseline (47.3%). This indicates that reducing or increasing the budget has only marginal influence in these settings. In contrast, in LONGGEN-BENCH (long-output), using the same 5% budget leads to substantial performance degradation in Quest (*e.g.*, 60.8%→17.6% on GSM8K). Notably, increasing the budget to 10%, 15%, and 20%, improves the accuracy to 32.3%, 50.9%, and 56.0%, respectively. This trend is because either missing or supplementing KV cache in long generation repeats over far more decoding steps, amplifying their influences (see Appendix B for full discussion).

Motivated by this observation, RetroAttention aims to efficiently supplement the KV cache by exploiting KV entries loaded by adjacent Queries in a local window; specifically, KV entries that are loaded by the *current* Query but were not available to *previous* Queries, as illustrated in Figure 1(a). However, this idea introduces two critical questions: (1) Are the currently loaded KV entries actually useful for improving the attention outputs of past Queries? and (2) How much informational benefit can we expect from leveraging such retrospective refinement?

**Usefulness of Loaded KV Entries for Past Queries.** Consecutive tokens often exhibit strong semantic correlations (Dai et al., 2024; Wu et al., 2024), suggesting that KV entries retrieved for the current Query may also be informative for nearby past Queries. To evaluate this, we compute the rank of currently loaded KV entries in past Queries. As shown in Figure 2(b), about 70–80% of current KV entries were previously within the top-$k$ set (navy bar), while 20–30% were not selected at all. Notably, the second-largest group—about 15–20%—are ranked within the next-$k$

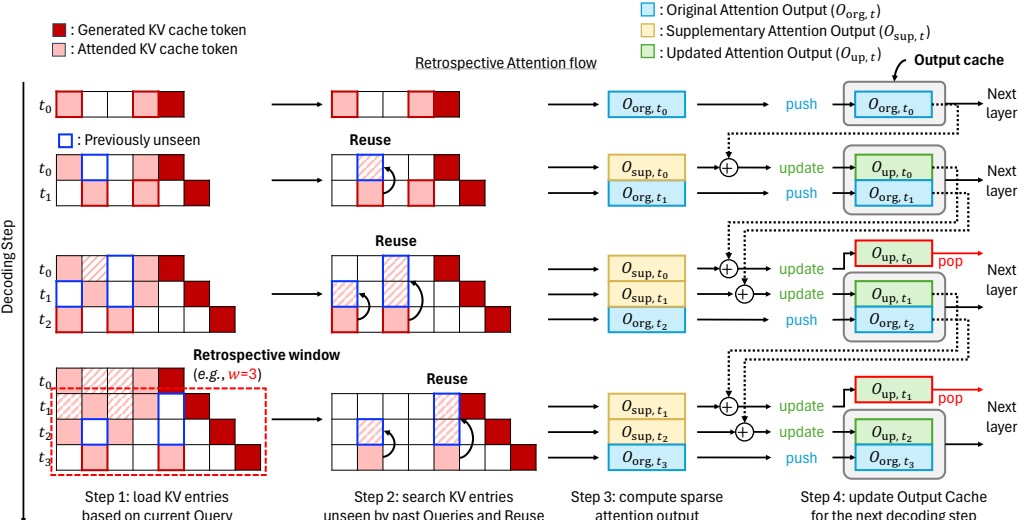

Figure 3: **Retrospective attention output update**. Overview of RetroAttention across three decoding steps, illustrating how previously unseen KV entries are retrospectively reused to refine past attention outputs via supplementary computation and output cache updates. This illustration assumes a retrospective window size of $w$=3.

range (orange bar), implying that they are still highly relevant yet were omitted in the past. This highlights the potential of currently loaded KV entries to supplement semantically meaningful but ignored content in the original top-$k$ selection (see Appendix B for results on other datasets).

**Informational Benefit of Retrospective Updates.** Retrospective updates enable past Queries to access KV entries beyond their original top-$k$ set. As decoding progresses, each past attention can be revised multiple times in subsequent decoding steps, leading to a cumulative expansion of supplemented KV entries. To quantify the total set of KV entries exposed to a Query over time—referred to as the *effective KV cache budget*—we count how many KV entries loaded by future Queries were previously unseen by a given Query (Figure 2(c)). The effective budget is computed as the union of all KV entries seen by a Query, excluding duplicates. Even with a single future Query ($n$=1), where each attention updates only its immediate predecessor, the effective budget increases by 1.17×. As more future Queries supplement, the effective budget continues to grow, reaching up to 1.60× at $n$=7. Notably, this benefit is achieved without increasing the actual KV cache budget.

**Relationship to Cache Selection Reuse.** Recent works (Yang et al., 2025; Wu et al., 2024) report that adjacent Queries tend to retrieve highly similar top-$k$ pages, motivating the reuse of previously selected pages. We observe the same trend in our setting (Figure 2(b)). However, in long-generation tasks, the impact of missing pages persists and accumulates over extended decoding. For example, integrating top-$k$ selection reuse into Quest reduces mean accuracy from 52.6% to 25.2% on GSM8K and from 56.8% to 29.2% on CSQA in LONGGENBENCH (Table 4 in Appendix B). These results suggest that reuse alone does not adequately preserve performance in long-generation settings, motivating mechanisms that mitigate error accumulation.

## 2.3 RETROSPECTIVE ATTENTION OUTPUT UPDATE

Building on insights from our motivational study, we hypothesize that currently loaded KV entries can effectively compensate for attention errors made in earlier steps. These errors stem from the lack of relevant Keys and Values in the attention output ($O$) computation. To correct this, our objective is to revise the previously computed $O$ vector in a retrospective manner. We introduce two key components to implement this idea: (1) supplementary attention output and (2) attention output cache. Figure 3 provides an overview of the retrospective attention update process.

**Supplementary Attention Output.** As illustrated in Steps 2 and 3 of Figure 3, RetroAttention computes attention outputs not only for the current decoding step but also retrospectively for pre-

vious steps. For the current step, attention is computed using sparse attention methods *e.g.*, Quest, yielding the output $O_{\text{org},t}$ (blue box). For past steps, RetroAttention computes attention between previous Queries and the currently loaded—but previously unseen—KV entries, producing $O_{\text{sup},t}$ (yellow box). We refer to this additional attention output as the *supplementary attention output*. While omitted in the figure, we maintain a mask that tracks the most recent decoding step in which each KV page was loaded, allowing us to identify previously unseen entries. Formally, the attention outputs are defined as follows:

$$O_{\text{org}, t} = \frac{\sum_{j \in S_t} \sum_{l \in \text{Page}_j} e^{Q_t K_l^\top / \sqrt{d}} V_l}{\sum_{i \in S_t} \sum_{l \in \text{Page}_i} e^{Q_t K_l^\top / \sqrt{d}}} \qquad O_{\text{sup}, t}^{t+s} = \frac{\sum_{j \in S_{t+s} \setminus \cup_{m=t}^{t+s-1} S_m} \sum_{l \in \text{Page}_j} e^{Q_t K_l^\top / \sqrt{d}} V_l}{\sum_{i \in S_{t+s} \setminus \cup_{m=t}^{t+s-1} S_m} \sum_{l \in \text{Page}_i} e^{Q_t K_l^\top / \sqrt{d}}} \qquad (2)$$

where $Q_t$ is the $t$-th Query vector, $S_t$ denotes the set of top-$k$ page indices loaded by $Q_t$, and $\text{Page}_i$ is the set of token indices in $i$-th page. $O_{\text{sup},t}^{t+s}$ refers to supplementary output generated at the $t+s$-th decoding step ($0 < s < w$) for $Q_t$. As described in Figure 3, the supplementation is not limited to adjacent steps. It is rather applied across multiple past Queries within a retrospective window $w$.

By combining $O_{\text{org},t}$ and $O_{\text{sup},t}^{t+s}$ for all $s$, we can extend the attention computation for the past Query beyond its original KV cache budget. Accordingly, our goal is to update $O_{\text{org},t}$ to $\text{softmax}(Q_t K^\top / \sqrt{d})V$, computed over $K, V$ in $\text{Page}_i$ where $i \in \cup_{0 \leq s < w} S_{t+s}$.

Inspired by the aggregation of partial attention results in FlashAttention (Dao et al., 2022), the $\text{softmax}$ equation can be rewritten to a linear combination of the original and supplementary attention outputs (see details in Appendix C), which we refer to as $O_{\text{up}, t}$. This allows us to *update* earlier outputs using newly available KV entries, efficiently correcting their errors. Discussions on the policy for *unseen* page detection appear in Appendix C.2 (*Retrospective Mask* paragraph). The merging of original and supplementary attention outputs is detailed in Appendix C.2, and their schedule is described in Appendix C.1 (Algorithm 1).

**Attention Output Cache.** Retrospective updates require access to the previous attention outputs of past Queries, which are typically discarded after their decoding steps. Recomputing these outputs would necessitate reloading the top-$k$ KV pages for each past Query, increasing the required KV cache budget in proportion to the retrospective window size, which is undesirable. To avoid this, we introduce an *attention output cache*, which stores multiple attention outputs (current and previous steps). This cache consumes marginal memory, with a size of $(w-1, B, L, D)$—corresponding to the restrospective window size $w$, batch size $B$, number of layers $L$, and hidden dimension $D$—and is independent of the generation length, unlike KV cache.

The cache operates via three actions, as depicted in Step 4 of Figure 3: (1) Push: attention output computed at each decoding step is stored in the cache. (2) Update: when supplementary attention outputs become available, the cached outputs, either the original $O_{\text{org}}$ or previously updated $O_{\text{up}}$, are reused to compute revised outputs via weighted combination. This allows both initial updates ($O_{\text{org}} \to O_{\text{up}}$) and re-updates ($O_{\text{up}} \to O_{\text{up}}$). (3) Pop: when the number of cached entries exceeds the window size, the oldest entry is evicted.

## 2.4 RETROSPECTIVE KV CACHE UPDATE

The above components describe how attention is updated within a single layer. However, they do not fully describe how refined past outputs propagate across layers and influence final generation. Here, we show how retrospective updates modify the KV cache in deeper layers.

**Influences on Deeper Layers.** As shown in Figure 4, the output cache from the layer $l$ provides multiple embeddings (*e.g.*, $O_{\text{up}, t_{1-2}}(l)$ and $O_{\text{org}, t_3}(l)$) to the layer $l + 1$. For the most recent step $t_3$, the layer $l + 1$ has not yet been computed. Thus, the projection layers generate Query $Q_{t_3}(l + 1)$, Key $K_{t_3}(l + 1)$, and Value $V_{t_3}(l + 1)$ for the first time. These Key and Value are newly appended to the KV cache (blue line). For previous steps $t_{1-2}$, we re-embed $O_{\text{up}, t_{1-2}}(l)$ to compute new Queries $\hat{Q}_{t_{1-2}}(l + 1)$, Keys $\hat{K}_{t_{1-2}}(l + 1)$, and Values $\hat{V}_{t_{1-2}}(l + 1)$. These differ from the original KV entries generated using $O_{\text{org}, t_{1-2}}(l)$ (*i.e.*, $K_{t_{1-2}}(l + 1)$ and $V_{t_{1-2}}(l + 1)$), which are now outdated. Hence, we overwrite the previous KV entries with updated ones: $K_{t_{1-2}}(l + 1) \to \hat{K}_{t_{1-2}}(l + 1)$ and $V_{t_{1-2}}(l + 1) \to \hat{V}_{t_{1-2}}(l + 1)$ (green line).

Once the updated Key-Value for $t_1$ to $t_2$ are overwritten in the KV cache, all subsequent decoding steps ($t > t_3$) automatically leverage these changes during attention. No special logic is required for deeper layers; they read from the KV cache as usual. This process naturally leads to a progressive reduction in attention errors across layers, as later queries attend to higher-quality representations of the past tokens.

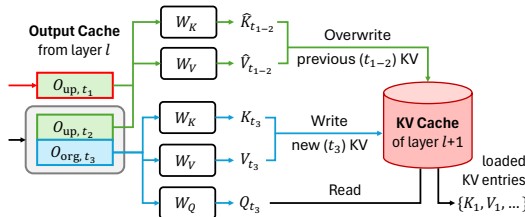

Figure 4: **Retrospective KV cache update**. Output cache from the previous layer provides multiple embeddings ($t_{1-3}$) to the current layer. The most recent output ($t_3$) is used to generate new KV entries, while earlier outputs ($t_{1-2}$) produce updated KV entries that overwrite previous ones in the current layer's KV cache. Feed-forward and normalization layers are omitted.

## 2.5 OVERHEAD ANALYSIS

Conventional attention operations heavily rely on GEMV (General Matrix-Vector multiplication) during decoding, leading to extremely low PE (Processing Element) utilization. We strategically design RetroAttention to exploit this idle parallelism for retrospectively updating past attention outputs and KV entries, using currently loaded KV entries. As a result, the additional computation in RetroAttention incurs only marginal overhead in memory and latency. To support this claim, we provide a per-layer overhead analysis.

**RetroAttention's Overhead in Attention Layers.** We analyze RetroAttention's Arithmetic Intensity (AI)—the ratio of FLOPs to bytes accessed from main memory—to show it operates in a memory-bound regime. Its FLOPs scale with the retrospective window size $w$, number of loaded KV pages $k_{\text{page}}$, Query heads $h_q$, tokens per page $P$, and head dimension $d$, and is estimated as $4w\,k_{\text{page}}\,h_q\,P\,d$ in FP16. This involves four types of memory access: (1) loading $2w\,h_q\,d$ bytes for Query vectors, (2) loading $4\,k_{\text{page}}\,h_k\,P\,d$ bytes for KV cache pages, (3) accessing $4\,h_k\,k_{\text{page}}$ bytes for a mask tracking KV page usage, and (4) writing $2w\,h_q\,d$ bytes for the output.

$$\text{AI}_{\text{attention}} = \frac{w\,k_{\text{page}}\,h_q\,P\,d}{w\,h_q\,d + k_{\text{page}}\,h_k\,P\,d + h_k k_{\text{page}}} \tag{3}$$

As $w\,h_q \ll k_{\text{page}}\,P$ and $h_k \ll P\,d$, Equation (3) simplifies approximately to $wh_q/h_k$. Given that modern GPUs typically shift to compute-bound above AI values of 200-400 (Zhu et al., 2025), RetroAttention remains memory-bound for moderate window sizes (*e.g.*, $w{<}100$), making latency primarily memory-driven. That is, we can quantify the latency overhead by comparing memory traffic. Baseline (*e.g.*, Quest) sparse attention loads $h_q d$ for Queries and $k_{\text{page}}\,h_k\,P\,d$ for KV cache pages. The memory traffic ratio is:

$$\frac{\text{Our Mem I/O}}{\text{Baseline Mem I/O}} = \frac{w\,h_q\,d + k_{\text{page}}\,h_k\,P\,d + h_k k_{\text{page}}}{h_q\,d + k_{\text{page}}\,h_k\,P\,d} \tag{4}$$

As KV loads ($k_{\text{page}}\,h_k\,P\,d$) dominate both numerator and denominator, our memory communication overhead is negligible (*i.e.*, the ratio stays near 1 for moderate $w$). This indicates marginal latency overhead from RetroAttention.

**RetroAttention's Overhead in Linear Layers.** Linear layers in RetroAttention are accompanied by $w$ embeddings, processing input matrices of shape $wb \times D$, where $b$ is a batch size and $D$ is the feature dimension. The AI of RetroAttention's linear layers is as follows:

$$\text{AI}_{\text{linear}} = \frac{w\,bD_{\text{in}}D_{\text{out}}}{w\,b(D_{\text{in}} + D_{\text{out}}) + D_{\text{in}}D_{\text{out}}}, \tag{5}$$

where $D_{\text{in}}$ and $D_{\text{out}}$ are the input and output dimensions of a linear layer. They remain memory-bound as long as $wb$ is below a few hundred, with overhead scaling with memory traffic rather than computation. Our method increases memory traffic (denominator in (5)) in the first term by $w$ times, but the second term ($D_{\text{in}}D_{\text{out}}$) dominates, yielding only minor latency overhead. See Appendix D for details.

Table 1: **Comparison with SOTA methods.** Accuracy results on GSM8K, MMLU and CSQA in LONGGENBENCH with three $n$ (*i.e.*, the number of questions in a prompt) values are summarized. All methods use relative KV cache budget of 0.15, which dynamically changes the budget to context length $\times$ 0.15 as decoding continues, with the minimum budget of 256 tokens. We set the retrospective window size ($w$) to 2.

| Method / Benchmark | $n$-question **GSM8K** ($\uparrow$) | | | | $n$-question **MMLU** ($\uparrow$) | | | | $n$-question **CSQA** ($\uparrow$) | | | |
|---|---|---|---|---|---|---|---|---|---|---|---|---|
| | 15 | 30 | 45 | Mean Acc. | 15 | 30 | 45 | Mean Acc. | 15 | 30 | 45 | Mean Acc. |
| Full Attention | 66.7 | 60.8 | 58.0 | 61.8 | 62.5 | 58.7 | 57.1 | 59.4 | 73.5 | 74.1 | 71.9 | 73.2 |
| StreamingLLM | 0.0 | 0.0 | 0.0 | 0.0 | 1.2 | 2.1 | 2.5 | 1.9 | 0.8 | 1.1 | 4.0 | 2.0 |
| TOVA | 0.2 | 0.3 | 0.2 | 0.2 | 9.7 | 8.7 | 9.4 | 9.3 | 6.7 | 6.3 | 11.1 | 8.1 |
| Quest | 58.2 | 50.9 | 48.6 | 52.6 | 58.8 | 54.9 | 50.6 | 54.8 | **70.9** | 53.5 | 46.1 | 56.8 |
| **RetroAttention (Ours)** | **61.3** | **52.6** | **55.4** | **56.5** | **59.3** | **55.4** | **51.2** | **55.3** | 68.8 | **59.1** | **53.0** | **60.3** |
| $\triangle$ Ours - Quest | +3.1 | +1.7 | +6.8 | +3.9 | +0.5 | +0.5 | +0.6 | +0.5 | -2.1 | +5.7 | +6.9 | +3.5 |

## 3 EXPERIMENTAL RESULTS

### 3.1 EXPERIMENTAL SETUP

**Datasets and Implementation Details.** Our main benchmark is LONGGENBENCH (Liu et al., 2024c) in which multiple reasoning tasks are sequentially concatenated into a single input prompt (details in Appendix F). Such prompt scheme is applied to CSQA (Talmor et al., 2019), MMLU (Hendrycks et al., 2021), and GSM8K (Cobbe et al., 2021) datasets, and a parameter $n$ determines the number of questions in a prompt. We follow the prompt construction and greedy decoding scheme according to the official repository[1]. Reasoning-intensive benchmarks such as AIME 2024, GPQA-DIAMOND (Rein et al., 2024), and LIVECODEBENCH (Jain et al., 2024) are included. We also include PG-19 (Rae et al., 2019) benchmark to estimate long-sequence language modeling performance using perplexity. Experiments are mostly performed with LLAMA-3.1-8B-INSTRUCT but with DEEPSEEK-R1-DISTILL-LLAMA-8B for the reasoning-intensive tasks. Our implementation adopts Quest's page selector (Tang et al., 2024) and is built on FlashInfer (Ye et al., 2025), a highly optimized implementation of attention.

**Baselines.** We compare three recent KV-cache compression methods—StreamingLLM (Xiao et al., 2023), TOVA (Wu et al., 2024), and Quest (Tang et al., 2024)—each employing a distinct token-selection strategy. StreamingLLM and TOVA are both eviction-based techniques. StreamingLLM relies on a static cache window whereas TOVA adaptively retains KV entries based on heuristic token significance estimation. We include these methods as they have not been rigorously evaluated under long-generation scenarios. Our primary baseline is Quest, a SOTA non-eviction method that selects a subset of tokens through Query-aware, paged sparse attention.

### 3.2 COMPARISON WITH SOTA METHODS

We compare the proposed method with the baselines in long-generation GSM8K, MMLU, and CSQA datasets, provided in LONGGENBENCH. Table 1 summarizes accuracy results on the three datasets as the number of questions concatenated in a prompt ($n$) increases. Overall input and output tokens range between about 1.3k-5.7k and 0.6k-4.2k, respectively. A higher $n$ indicates longer input and output lengths. See data statistics in Appendix F.

**Accuracy Analysis for Eviction-based Baselines.** StreamingLLM and TOVA fail across all three datasets, even when $n$ is small. This suggests that their failure stems from an inherent limitation of the methods in sequential question-answer scenarios, rather than a challenge specific to *long* generation. Both approaches permanently evict KV cache entries, meaning that tokens relevant to later questions may be discarded during earlier decoding steps and can never be recovered. Thus, the model is unable to attend to upcoming content when needed, revealing a critical weakness of eviction-based strategies.

**Accuracy Analysis for Non-eviction Baseline.** While exhibiting higher accuracy, Quest also experiences a noticeable accuracy drop at high $n$ (see Table 1). In particular, the performance gap between full cache and Quest becomes larger as $n$ grows (*e.g.*, -2.6%p, -20.6%p, and -25.8%p in

---

[1] https://github.com/Dominic789654/LongGenBench

Table 2: **Comparison in larger models.** Accuracy results in QWEN2.5-32B-INSTRUCT and QWEN2.5-14B-INSTRUCT models. Experiment settings are identical with Table 1.

| Method / Benchmark | $n$-question **GSM8K** ($\uparrow$) | | | | $n$-question **MMLU** ($\uparrow$) | | | | $n$-question **CSQA** ($\uparrow$) | | | |
|---|---|---|---|---|---|---|---|---|---|---|---|---|
| | 15 | 30 | 45 | Mean Acc. | 15 | 30 | 45 | Mean Acc. | 15 | 30 | 45 | Mean Acc. |
| | | | | | QWEN2.5-32B-INSTRUCT | | | | | | | |
| Full Attention | 92.1 | 91.7 | 90.8 | 91.5 | 81.8 | 81.0 | 81.2 | 81.3 | 86.1 | 87.0 | 87.1 | 86.7 |
| Quest | 89.2 | 78.8 | 83.3 | 83.8 | 81.0 | 79.4 | 79.5 | 80.0 | 82.3 | 69.3 | 74.4 | 75.4 |
| **RetroAttention (Ours)** | 89.8 | 89.1 | 84.4 | 87.8 | 80.9 | 80.3 | 79.9 | 80.4 | 85.3 | 75.0 | 78.7 | 79.7 |
| | | | | | QWEN2.5-14B-INSTRUCT | | | | | | | |
| Full Attention | 88.7 | 84.4 | 76.6 | 83.2 | 78.2 | 76.6 | 73.6 | 76.1 | 83.7 | 83.5 | 83.5 | 83.6 |
| Quest | 79.0 | 76.5 | 64.1 | 73.2 | 76.0 | 73.0 | 71.1 | 73.3 | 69.4 | 46.8 | 58.1 | 58.1 |
| **RetroAttention (Ours)** | 83.1 | 80.7 | 69.1 | 77.6 | 75.8 | 73.7 | 71.0 | 73.5 | 73.3 | 59.5 | 63.6 | 65.5 |

Figure 5: **Ablation studies**. (a) Comparison between full cache (no compression), Quest, and ours in LONGGENBENCH with three retrospective window sizes. We set the relative budget to 0.15. (b)-(d) present the trade-off between the attention memory traffic and accuracy with three relative budget ($b$) values in CSQA, MMLU, and GSM8K, respectively. We set $n$ to 30.

CSQA), implying the cumulative errors caused by approximated attention. In contrast, our approach is robust to such performance degradation, consistently outperforming Quest in nearly all configurations. The performance gain of RetroAttention over Quest is particularly large with higher $n$ (*e.g.*, 6.8%p in GSM8K, 6.9%p in CSQA), highlighting the effectiveness of our retrospective updates. Comparison with other non-eviction baselines is also discussed in Appendix F.3.

**Accuracy Improvements in Larger Models.** RetroAttention also delivers consistent gains in larger models (>8B). Table 2 reports results on LONGGENBENCH using QWEN2.5-32B-INSTRUCT and QWEN2.5-14B-INSTRUCT. For the 14B model, RetroAttention achieves an average improvement over Quest of +7.4% on CSQA, +4.4% on GSM8K, and +0.2% on MMLU. For the 32B model, the corresponding gains are +4.3% (CSQA), +4.0% (GSM8K), and +0.4% (MMLU). This trend mirrors the results in Table 1, further demonstrating the generality and scalability of RetroAttention in larger models.

We also compare our method against Quest with larger retrospective window sizes ($w$). Figure 5(a) shows accuracy results of full cache, Quest, and RetroAttention under the three window sizes ($w \in \{2, 4, 8\}$). As $w$ increases, RetroAttention approaches the performance of the full cache baseline more closely across all benchmarks.

## 3.3 COMPUTATIONAL EFFICIENCY

**Memory Analysis.** Increasing the KV budget is a straightforward way to improve the accuracy at the cost of additional memory traffic. Our method redesigns such trade-off through retrospective updates. Figures 5(b)–(d) show comprehensive comparisons between Quest and ours with three relative KV cache budgets ($b$). The relative budget adaptively changes the amount of KV entries to be loaded by context length $\times b$. As shown in the figure, RetroAttention yields significant accuracy gains over Quest at a given budget with negligible memory traffic overhead of 3.0% in CSQA, 1.6% in MMLU, and 2.0% in GSM8K on average.

**End-to-End Latency.** Figure 6(a) presents the average end-to-end latency per decoding step for the full-cache baseline, Quest, and ours (setup details in Appendix F). RetroAttention consistently

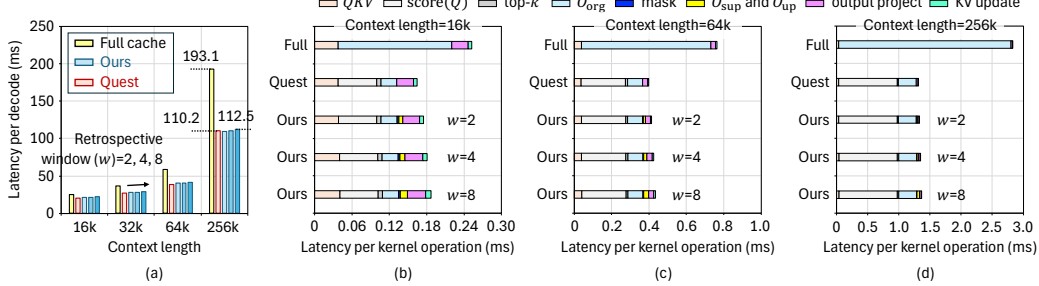

Figure 6: **Latency analysis**. (a) Comparison of end-to-end latency per decoding between full cache (no compression), Quest, and ours with three retrospective window sizes. (b)-(d) Comparison of kernel-level latency breakdown between full cache (denoted as Full), Quest, and ours with three context lengths. Latency results are estimated and then averaged in the 1024 decoding steps followed by prefill with context length. We set the relative budget ($b$) to 0.15.

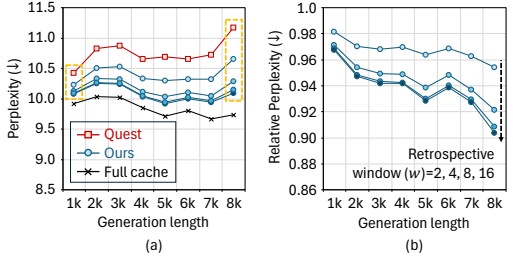

Figure 7: **Comparison in language modeling**. The average Perplexity results of the full-cache, Quest, and ours in PG-19 are compared. First 8k tokens are prefilled with full KV, followed by 8k sparse decoding. Relative Perplexity denotes a ratio between Quest's and ours. We set $b$ to 0.15.

Table 3: **Comparison in reasoning-intensive tasks.** Accuracy results on AIME 2024, GPQA-DIAMOND and LIVECODEBENCH-V5 datasets are summarized with four retrospective window sizes. We set $b$ and $w$ to 0.15 and 2.

| Method / Benchmark | AIME24 Pass@1 (↑) | GPQA-D Pass@1 (↑) | LCBv5 Pass@1 (↑) |
|---|---|---|---|
| Full Attention | 47.1 | 38.9 | 37.6 |
| StreamingLLM | 0.4 | 19.7 | 1.5 |
| TOVA | 0.4 | 0.0 | 7.0 |
| Quest | 33.8 | **33.6** | 32.7 |
| **RetroAttention (Ours)** | **39.2** | 33.6 | **34.1** |
| Δ Ours - Quest | +5.4 | 0.0 | +1.4 |

outperforms the full-cache baseline and remains at similar latency with Quest across all context lengths. The additional latency of RetroAttention over Quest is minimal, less than 1ms per token for $w$=2 and around 2ms for $w$=8. Importantly, this overhead remains effectively constant regardless of context length or retrieved cache size, aligned with our theoretical analysis in Section 2.5, which attributes the overhead primarily to additional memory I/O.

**Kernel-level Attention Latency.** Figures 6(b)–(d) show the average kernel-level latency involved in the attention operation. The latency for scoring each KV page ($\mathrm{score}(Q)$) and top-$k$ selection remain unchanged from Quest to RetroAttention, as they operate only on the most recent Query. Additional operations introduced by RetroAttention—such as projecting historical Queries and Keys, overwriting the KV cache, managing masks, and merging attention outputs—incur overhead. However, these costs are small and largely constant across context lengths, making their relative impact increasingly negligible as the context grows.

## 3.4 COMPARISON IN OTHER TASKS

**Reasoning Tasks.** Table 3 presents accuracy results on AIME 2024, GPQA-DIAMOND, and LIVECODEBENCH-V5. Similarly in LONGGENBENCH, the eviction-based methods suffer large degradation compared with full attention; for example, TOVA often fails to follow instructed format during decoding in GPQA-DIAMOND. RetroAttention exhibits similar accuracy in GPQA-D yet significant gains over Quest are observed in the others.

**Long-context Input.** While our method is not designed for long-context inputs, we also evaluate RetroAttention on LONGBENCH (Bai et al., 2023), a benchmark for long-context inputs (see Appendix G). The evaluated categories include Single-doc and Multi-doc QA, Summarization, Code, and Few-shot tasks. RetroAttention achieves comparable performance to Quest, likely because the examples in LONGBENCH are mostly short-generation (<100 tokens), where the benefits of retrospective updates are less pronounced.

**Language Modeling.** Figure 7 compares Perplexity on PG-19 measured at 1k-token intervals during decoding. Our method consistently achieves lower Perplexity across all intervals than Quest (Figure 7(b)), with particularly strong improvements in later stages of generation. This verifies that the retrospective updates reliably enhance output logits across different window sizes ($w$). We note that the gains saturate beyond $w=8$, implying an upper bound in improvements from RetroAttention.

**Performance Gain Saturation.** A larger retrospective window increases the effective KV budget. Hence, the observed performance saturation in RetroAttention can be understood by asking why increasing the effective budget does not yield continuous accuracy improvements. As shown in Appendix C.3, most of the informative KV elements (those with high attention weights) are included first. As the window size $w$ increases, the additional KV elements become progressively less important. Their marginal contribution to accuracy becomes smaller, which explains why the gains diminish and eventually saturate beyond $w=8$. Importantly, this behavior is not specific to RetroAttention. Even when increasing the actual KV cache size, accuracy improvements also plateau once the model has already incorporated the dominant, high-importance tokens (see Figures 5(c)–(d)).

## 4 RELATED WORKS

**Eviction-based KV Cache Compression.** Several KV cache eviction strategies have been proposed to reduce memory usage and accelerate attention computation. Methods such as StreamingLLM (Xiao et al., 2023) retain only the earliest and most recent tokens, while Fast-Gen (Ge et al., 2023) and SnapKV (Li et al., 2024) evicts low-importance tokens after the prefill stage assuming stable attention patterns. Dynamic approaches such as H2O (Zhang et al., 2023), KeyFormer (Adnan et al., 2024), and TOVA (Oren et al., 2024) adaptively determine which tokens to retain based on historical attention statistics. However, all these methods permanently discard selected KV entries, overlooking possible later relevance of evicted tokens—causing cumulative errors or missed dependencies.

**Non-eviction KV Cache Compression.** Non-eviction methods preserve the full KV cache but reduce computation by sparsely loading only relevant tokens per decoding step, allowing revisitation of previously irrelevant entries. Pioneering works, Quest (Tang et al., 2024) and ArkVale (Chen et al., 2024a) identify top-$k$ relevant pages of KV cache (Kwon et al., 2023) via similarity between Queries and page-level summaries. InfLLM (Xiao et al., 2024), SqueezedAttention (Hooper et al., 2024), and ClusterKV (Liu et al., 2024b) use representative or centroid Key vectors to generate these summaries. Other methods such as RetrievalAttention (Liu et al., 2024a) and TokenSelect (Wu et al., 2024) dynamically select relevant cache subsets at token granularity. While effective for long-context inputs, these approaches still overlook cumulative attention errors during generation.

**KV Cache Compression for Long Generation.** Few studies address KV cache compression for long generation. Song et al. (2025) exploit the semantic sparsity of reasoning traces by periodically evicting low-importance entries, but may inherit limitations common to eviction-based methods, especially under evolving attention patterns. Concurrent with our work, Sun et al. (2025) raise similar concern about the cumulative errors caused by sparse attention and propose periodically refreshing previous attention outputs using dense attention (*i.e.*, full KV cache). However, this incurs significant computational overhead due to repeated dense attention, in contrast to our contributions—reducing the errors without additional KV cache budget.

## 5 CONCLUSION

We introduce RetroAttention, a lightweight yet effective KV cache compression technique, to mitigate cumulative attention errors in long-context generation. Our approach fundamentally differs from conventional sparse attention. Previous methods treat attention output as fixed: Once computed, the loaded KV entries are discarded and the errors persist throughout the model. RetroAttention instead reuses those KV entries to retrospectively revise past outputs and overwrite stale KV states, enabling consistent correction of cumulative attention errors—without increasing KV cache budget. This design yields consistent accuracy improvements in long-context generation, while maintaining negligible memory and latency overhead.

ETHICS STATEMENT

Our work seeks to improve the efficiency and accuracy of long-context inference in large language models by proposing a new KV cache update mechanism. While language models inherently raise concerns around potential misuse, bias, and fairness, our contributions are confined to algorithmic and efficiency-focused improvements. We do not anticipate introducing risks beyond those already inherent in the underlying models.

Large Language Models (LLMs) were not involved in formulating research ideas, designing methods, or conducting analysis. Their use was strictly limited to polishing the manuscript, such as improving clarity, grammar, and style of sentences originally written by the authors. No scientific content, arguments, or experimental descriptions were generated by LLMs, and their role was confined to editorial assistance only.

REPRODUCIBILITY STATEMENT

This paper introduces novel method that retrospectively revises past attention outputs with new KV entries, enabling long-context models to correct accumulated approximation errors efficiently. To facilitate reproducibility, we provide the complete source code as a downloadable repository along with documentation detailing evaluation procedures. All benchmark datasets used in our experiments are publicly available, and we provide the scripts employed during evaluation to ensure consistency with our reported results. We provide further clarification and an extended analysis of our theoretical claims in the appendix. In particular, we elaborate on the underlying assumptions, present additional derivations that were omitted from the main text for brevity, and supplement our arguments with supporting empirical evidence where appropriate. These resources allow independent researchers to reliably replicate and validate our findings.

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

## A  OVERVIEW

This appendix provides supplementary material for our manuscript, "Retrospective Sparse Attention for Efficient Long-Context Generation." It includes additional implementation details and theoretical analysis that complement Section 2, and provides detailed descriptions of the motivation and experimental settings. Section B discusses the motivation behind our method in detail with additional experimental evidence. Section C describes additional implementation details of the main RetroAttention algorithm, including attention output updates and the retrospective mask logic. In Section D, extending the discussion in Section 2.5 of the main paper, we elaborate on the overhead analysis for linear layers, showing that the additional latency remains negligible due to their memory-bound nature. Section E quantifies RetroAttention's memory footprint in addition to the memory traffic discussion in Section 2.5. Section F outlines dataset statistics, prompt formats, and experimental configurations for our latency and accuracy evaluations. Lastly, Section G shows evaluation results on long-context input benchmarks.

## B  MOTIVATIONAL STUDY

In the motivational study (main paper), we discuss the difference between long-context *input* and *output* tasks with respect to their dependency on the KV budget. This motivates us to consistently supplement missing KV cache, efficiently within a small local window, as they will benefit across longer decoding steps and effectively mitigates the error accumulation. However, the differences of top-$k$ pages in adjacent Queries are often considered negligible (Yang et al., 2025), even showing little degradation when reusing top-$k$ pages from previous Queries in certain tasks. We further strengthen our argument through the following experimental evidence: (1) 20% of top-$k$ pages are not shared in adjacent Queries (in multiple datasets), and (2) such amount can make significant influences in long-generation tasks. These results highlight that though the amount of missed or supplemented KV cache is seemingly small, their influences can be large in long generation.

**How many top-$k$ pages are shared in adjacent Queries?**  We demonstrate that adjacent Queries share 82.8% (interval: 1) to 73.8% (interval: 8) top-$k$ pages in Figure 2(b). In Figure 8(a)-(c), we examine the token distribution across the three datasets, including GSM8K, CSQA, and MMLU. More than a quarter of top-$k$ pages *do not overlap* in the 8-step interval in common, which we do not consider little, particularly given that we can leverage them without increasing the actual KV budget. In addition, retrospective supplementation is influenced by all the past queries in the window (not like a single query influences only one), reaching minimum 17.2% ($w$=2; GSM8K) to maximum 69.8% ($w$=8; MMLU) increases in the effective KV budget. We confirm such trends hold across multiple datasets, including GSM8K, CSQA, and MMLU.

**Are the non-overlapping 20% of top-$k$ pages important?**  Methods based on reusing previously selected top-$k$ pages across multiple decoding steps have shown strong empirical performance, particularly in long-input benchmarks (*e.g.*, LONGBENCH, NIAH). Such approaches implicitly assume that the non-overlapping portion (approximately 20% of top-$k$ pages) has limited impact on downstream decoding. To examine this in long-generation settings, we integrate a reuse-based cache selection strategy into Quest and evaluate it on LONGGENBENCH. We observe substantial accuracy degradation even with a reuse interval of 2 decoding steps: mean accuracy decreases from 52.6% to 25.2% on GSM8K, from 54.8% to 34.7% on MMLU, and from 56.8% to 29.2% on CSQA (Top-$k$ Reuse ($f$=2) in Table 4). These results indicate that the non-overlapping portion of top-$k$ pages can influence decoding trajectories in long-generation tasks.

## C  IMPLEMENTATION DETAILS

### C.1  RETROATTENTION ALGORITHM FLOW

Algorithm 1 presents the detailed logic of RetroAttention. Lines 2, 5, 7, and 8 shows the additional operations introduced by RetroAttention: updating previous KV caches, updating the attended mask cache and building masks for the attention kernel, merging cached and current partial outputs, and caching current outputs. The attention operation (Line 6) processes $w$ Queries instead of a single

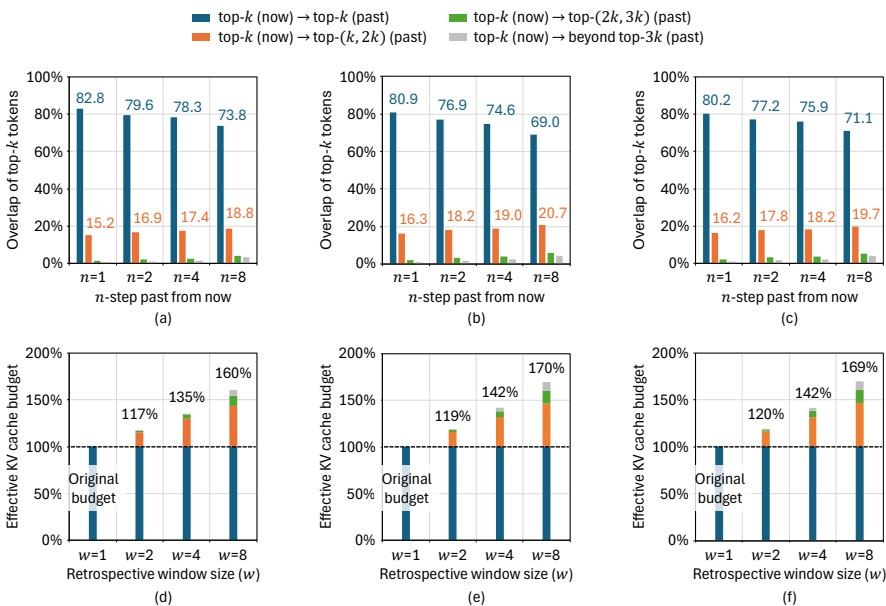

Figure 8: **Token distribution analysis in long generation**. (a)-(c) present the overlap of the current top-$k$ KV entries in the $t$-step prior decoding, categorized into different top-$k$ intervals, in GSM8K, CSQA, and MMLU datasets, respectively. (d)-(f) are the effective KV budget depending on the retrospective window sizes, in GSM8K, CSQA, and MMLU datasets, respectively.

Table 4: **Comparison against top-$k$ selection reuse**. The reuse-based strategy reuses previously loaded top-$k$ pages and is integrated into Quest. Here, $f$=2 indicates that top-$k$ pages are refreshed every two decoding steps. We set the relative budget $b$ to 0.15.

| Method / Benchmark | $n$-question **GSM8K** ($\uparrow$) | | | | $n$-question **MMLU** ($\uparrow$) | | | | $n$-question **CSQA** ($\uparrow$) | | | |
|---|---|---|---|---|---|---|---|---|---|---|---|---|
| | 15 | 30 | 45 | Mean Acc. | 15 | 30 | 45 | Mean Acc. | 15 | 30 | 45 | Mean Acc. |
| Full Attention | 66.7 | 60.8 | 58.0 | 61.8 | 62.5 | 58.7 | 57.1 | 59.4 | 73.5 | 74.1 | 71.9 | 73.2 |
| Quest | 58.2 | 50.9 | 48.6 | 52.6 | 58.8 | 54.9 | 50.6 | 54.8 | 70.9 | 53.5 | 46.1 | 56.8 |
| Top-$k$ Reuse ($f$=2) | 22.4 | 22.3 | 31.0 | 25.2 | 39.6 | 32.5 | 32.0 | 34.7 | 40.7 | 25.1 | 21.9 | 29.2 |
| RetroAttention ($w$=2) | 61.3 | 52.6 | 55.4 | 56.5 | 59.3 | 55.4 | 51.2 | 55.3 | 68.8 | 59.1 | 53.0 | 63.0 |

Query. As our method is not constrained by a specific top-$k$ page selection method (*e.g.*, Quest), we can integrate RetroAttention into any non-eviction dynamic sparse attention methods. To efficiently support these additional operations, we implement custom CUDA kernels—for page masking (Line 2) and incremental updates (Line 7)—and modify the main attention kernel to handle custom attention masking. We also extend the KV cache append kernel to support updates. Our modified attention kernel builds on FlashInfer (Ye et al., 2025), a high-performance attention computation framework.

## C.2 UPDATED ATTENTION OUTPUT

As discussed in Section 2.3, the updated attention output $O_{\text{up}, t}$ is computed by a weighted linear combination of the original and supplementary attention outputs. Let us rewrite $O_{\text{org}, t}$ and $O_{\text{sup}, t}$ in Equation 2 with explicit superscripts to clarify which decoding step each tensor comes from as follows:

$$O_{\text{org}, t} = \frac{\sum\limits_{j \in S_t} \sum\limits_{l \in \text{Page}_j} e^{Q_t^t K_l^{t\top}/\sqrt{d}} V_l^t}{\sum\limits_{i \in S_t} \sum\limits_{l \in \text{Page}_i} e^{Q_t^t K_l^{t\top}/\sqrt{d}}} \tag{6}$$

---

**Algorithm 1:** RETROATTENTION

---

**Input:** page selector PAGESELECT; queries $Q_{i:t}$, keys $K_{i:t}$, values $V_{i:t}$; K/V cache $C$; cached
    output $\hat{O}_{i:t-1}$; cached attention-lse $\hat{Z}_{i:t-1}$; window size $w$

**Output:** outputs $O_{i:t}$; updated cache $C$

1 $i \leftarrow t - w + 1$ ;               // Window start index
2 UpdateKV$(K_{i:t-1}, V_{i:t-1}, C)$;
3 AppendKV$(K_t, V_t, C)$;
4 $S_t \leftarrow$ PAGESELECT$(Q_t, C)$;
5 $M \leftarrow$ UpdateAndBuildMask$(S_t, C)$;
6 $(O_{i:t}, Z_{i:t}) \leftarrow$ Attention$(Q_{i:t}, S_t, M, C)$;
7 $(O_{i:t-1}, Z_{i:t-1}) \leftarrow$ MergeOutput$(\hat{O}_{i:t-1}, \hat{Z}_{i:t-1}, O_{i:t-1}, Z_{i:t-1})$;
8 CacheOutput$(O_{\max(i,t-w+2):t}, Z_{\max(i,t-w+2):t}, C)$;
9 **return** $O_{i:t}, C$

---

$$O^{t+s}_{\text{sup},\,t} = \frac{\displaystyle\sum_{j \in S_{t+s} \setminus \cup^{t+s-1}_{m=t} S_m} \sum_{l \in \text{Page}_j} e^{Q^{t+s}_t K^{t+s\top}_l / \sqrt{d}} V^{t+s}_l}{\displaystyle\sum_{i \in S_{t+s} \setminus \cup^{t+s-1}_{m=t} S_m} \sum_{l \in \text{Page}_i} e^{Q^{t+s}_t K^{t+s\top}_l / \sqrt{d}}} \tag{7}$$

where the superscripts $t$ and $t + s$ indicate the decoding step in which the tensor ($QKVO$) was computed and the subscript $t$ indicates its corresponding token index. As discussed in Section 2.4, retrospective updates to the hidden states of previous tokens propagate to subsequent layers, where the corresponding $QKV$ vectors are also updated. Therefore, it is necessary to distinguish $QKV$ vectors for the same token index using superscripts that indicate the time at which they were computed. For example, $Q_t$ is generated at the decoding step $t$ for the first time, but at $t + s$, the updated attention outputs from the previous layer provide new embeddings to the current layer and then generates another $Q_t$ (omitted in green line in Figure 4). To distinguish these two $Q_t$s, we denote the first $Q_t$ generated at $t$ as $Q^t_t$ and the second $Q_t$ generated at $t + s$ as $Q^{t+s}_t$.

As the first supplementary attention becomes available at step $t+1$ (*i.e.*, $s=1$), RetroAttention merges $O_{\text{org},t}$ and $O^{t+1}_{\text{sup},t}$ by a weighted average whose coefficients are the respective softmax normalizers:

$$\begin{aligned}
O^{t+1}_{\text{up},\,t} &= \frac{\alpha_{\text{org}} \cdot O_{\text{org},\,t} + \alpha^{t+1}_{\text{sup}} \cdot O^{t+1}_{\text{sup},\,t}}{\alpha_{\text{org}} + \alpha^{t+1}_{\text{sup}}} \\
\alpha^{t+1}_{\text{up}} &= \alpha_{\text{org}} + \alpha^{t+1}_{\text{sup}} \\
\alpha_{\text{org}} &= \sum_{i \in S_t} \sum_{j \in \text{Page}_i} e^{Q^t_t K^{t\top}_j / \sqrt{d}} \\
\alpha^{t+1}_{\text{sup}} &= \sum_{i \in S_{t+1} \setminus S_t} \sum_{j \in \text{Page}_i} e^{Q^{t+1}_t K^{t+1\top}_j / \sqrt{d}}
\end{aligned} \tag{8}$$

This procedure is applied recursively as additional supplementary outputs arrive—that is, $O^{t+2}_{\text{sup},t}$ and $O^{t+1}_{\text{up},t}$ are merged through $\alpha^{t+2}_{\text{sup}}$ and $\alpha^{t+1}_{\text{up}}$ by the same mechanism:

$$\begin{aligned}
O^{t+s+1}_{\text{up},\,t} &= \frac{\alpha^{t+s}_{\text{up}} \cdot O^{t+s}_{\text{up},\,t} + \alpha^{t+s+1}_{\text{sup}} \cdot O^{t+s+1}_{\text{sup},\,t}}{\alpha^{t+s}_{\text{up}} + \alpha^{t+s+1}_{\text{sup}}} \\
\alpha^{t+s+1}_{\text{up}} &= \alpha^{t+s}_{\text{up}} + \alpha^{t+s+1}_{\text{sup}} \\
\alpha^{t+s+1}_{\text{sup}} &= \sum_{i \in S_{t+s+1} \setminus \cup^s_{r=0} S_{t+r}} \sum_{j \in \text{Page}_i} e^{Q^{t+s+1}_t K^{t+s+1\top}_j / \sqrt{d}}
\end{aligned} \tag{9}$$

**Evolving Query Representations in Past Attention.** As discussed in the above paragraph, the updated attention output is transferred to the next layer and then projected into a new Query repre-

Table 5: **Relative attention-score gain by retrospective updates**. Across the three datasets of GSM8K, MMLU, and CSQA, the first update provides the largest recovery, while later updates contribute diminishing gains. We use few sampled data in LONGGENBENCH $n$-question prompts and set $n$=30, $w$=8, and $b$=0.15.

| Benchmark | Query indices in retrospective updates (current→past) | | | | | | | |
|---|---|---|---|---|---|---|---|---|
| | $t$=0→0 | $t$=1→0 | $t$=2→0 | $t$=3→0 | $t$=4→0 | $t$=5→0 | $t$=6→0 | $t$=7→0 |
| MMLU | 1.000 | 0.054 | 0.018 | 0.011 | 0.007 | 0.007 | 0.005 | 0.004 |
| GSM8K | 1.000 | 0.044 | 0.016 | 0.009 | 0.006 | 0.004 | 0.004 | 0.004 |
| CSQA | 1.000 | 0.650 | 0.028 | 0.015 | 0.013 | 0.010 | 0.010 | 0.009 |

sentation. This implies that Query representation at a given layer evolve as the update repeats. Dao et al. (2022) analytically show that the aggregation of two partial softmax outputs can be exactly same as the original softmax output. This assumes the partial outputs are generated at the same decoding steps (*i.e.*, based on the same Query representation), whereas RetroAttention merges the attention output asynchronously generated across a retrospective window. This leads to a mismatch in Query representations due to their evolution during retrospective updates.

Our updated attention output, merging multiple outputs across a retrospective window, is an approximation of the attention output on the total supplemented KV entries that would have been computed as normal sparse attention. We assume that the $QKV$ representations evolve smoothly across decoding steps, such that those from consecutive steps can be regarded as approximately equivalent. Empirically, the $\ell_2$ distance between the same vector across adjacent steps (*i.e.*, the change induced by one retrospective update) is typically less than 5% of its original $\ell_2$ norm for Query and Key vectors on average across various tasks, and less than 10% for Value vectors.

**Retrospective Mask.** The supplementary attention outputs are computed only on unseen KV cache pages. We introduce a *retrospective mask* ($\mathcal{A}$) to identify which pages have been already attended before and avoid their duplicated updates. Specifically, for each page $p$, we store $\mathcal{A}_h[p]$, the last decoding step in which the page was loaded for a head $h$. This indicates that the attention outputs preceding the $\mathcal{A}_h[p]$-th decoding step are already supplemented with the page $p$, if unseen. Accordingly, when computing the supplementary attention for the past (*e.g.*, decoding step $t$) Query and head $h$, we mask out any page $p$ such that $\mathcal{A}_h[p] \geq t$. Let $k_{\text{page}}$ be the number of loaded pages, $\mathcal{P}$ the global page pool (i.e., the entire KV cache), $L$ the sequence length, and $P$ the page size. This incremental masking logic relies only on tracking the latest Query per page, requiring $h_k |\mathcal{P}| = h_k L/P$ integers per layer for tracking and $h_k k_{\text{page}}$ integers to be loaded per kernel call, assuming $h_k$-wise page selection. Algorithm 2 describe the detailed mask management logic.

## C.3 SUPPLEMENTED ATTENTION WEIGHTS

This section provides empirical evidence on *why performance gain saturates in RetroAttention* discussed in Section 3.4. We suggest that such saturation is attributed from that the most informative KV elements (high attention weights) are selected first, and the supplemented ones become progressively less important (low attention scores), as the retrospective update progresses.

To demonstrate this, we measured the relative attention-score gain contributed by each update. For example, in Table 5 ($w$=8 setting), the first column represents the attention scores, *i.e.*, $\text{sum}(\exp(QK/\sqrt{d}))$, at $t$=0 Query (its original attention), and the next columns represent the attention scores supplemented from the next $t$=1, $t$=2, ... , $t$=7 Queries to this $t$=0 Query. Each supplementation is normalized by its original attention scores.

Empirically, the first retrospective update recovers 4–7% of the baseline attention mass, but subsequent updates contribute progressively less, indicating that later updates add only marginal signal. This is because attention weights are heavily concentrated on a small subset of tokens, expanding the retrospective window yields a non-linear and diminishing recovery of full-cache information. This directly explains the observed saturation in performance.

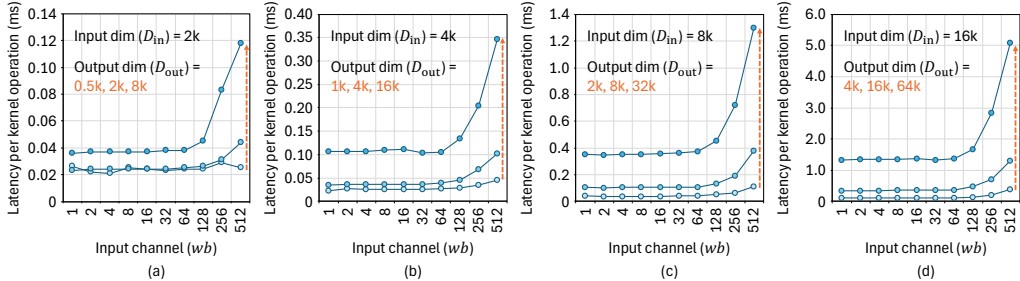

Figure 9: **Latency analysis for linear layers**. Kernel latency of RetroAttention in a linear layer with various input and output dimensions (denoted in the figure) and input channel sizes. For RetroAttention, the input channel size equals $w \times b$, where $w$ is the retrospective window size and $b$ is the batch size.

## C.4 SENSITIVITY TO RETROSPECTIVE WINDOW

The trade-off between accuracy and computation is essentially controlled by the retrospective window size $w$, the only control parameter in RetroAttention. The optimal trade-off would be achieved when accuracy gain is significant (or saturated) while computation (particularly latency) overhead is still marginal.

**Accuracy.** The accuracy is most sensitive when moving from $w$=0 to small windows ($w$=2 and 4); beyond that, gains quickly exhibit diminishing returns and largely saturate around $w$=8 (as also seen in Figure 5 and Figure 7). This aligns with our attention-mass analysis given in the answer above.

**Computation.** As described in Section 2.5 (Overload Analysis), the overhead scales roughly linearly with w, as each additional update adds extra query loads and FFN activations. However, this cost remains small, as the additional computation does not significantly contribute to the latency in memory-bound regimes.

This mismatch in how accuracy and overhead scale determines the practical trade-off. In typical long-generation scenarios, where decoding is strongly memory-bound, using a larger window (*e.g.*, $w$=8) is generally preferred: it provides near-saturated accuracy gains while the added latency remains minimal due to the bandwidth-bound regime. However, in settings where latency is more critical or the workload is not strictly memory-bound, practitioners may choose smaller windows ($w$=2 or 4) to further reduce overhead while still recovering a substantial portion of the accuracy benefits. Thus, RetroAttention offers a knob that can be tuned depending on the runtime constraints.

## D LINEAR LAYER OVERHEAD ANALYSIS

For an input of shape $n \times D_{\text{in}}$ and a projection weight of shape $D_{\text{in}} \times D_{\text{out}}$, the arithmetic intensity (AI) of linear layers in FP16 precision is given:

$$\text{AI}_{\text{linear}} = \frac{nD_{\text{in}}D_{\text{out}}}{nD_{\text{in}} + D_{\text{in}}D_{\text{out}} + nD_{\text{out}}} \tag{10}$$

where $n$ is batch $\times$ sequence length. This shows that when $n \ll D_{out}$ (which is typical in decoding stage), the AI becomes small, indicating that the operation is *memory-bound* rather than compute-bound. This allows models to have capacity for more computation with only negligible latency overhead. Figure 9 shows that the linear layer latency remains nearly constant until $n$ exceeds a certain threshold in diverse $D_{in}$ and $D_{out}$ combinations. These observations validate that the overhead per decoding step introduced by RetroAttention—whose linear layers process $w$ times larger input per step (*i.e.*, $n$ is $w \times$ batch)—is practically negligible unless $n$ is around 128-256. Moreover, this overhead is independent of sequence length and the number of loaded KV entries, making it suitable for long generation scenarios.

**Batch-size impacts on decoding throughput.** We report decoding-throughput measurements with and without RetroAttention across a range of batch sizes and long-context settings (see Table 6). Because recent models (*e.g.*, reasoning-oriented) often generate more than 10k tokens, we

Table 6: **Batch-size impacts on throughput**. Experiments are performed in Llama3.1-8B-Instruct model using a single A100 80GB GPU. Throughput is estimated by estimating the average decoding latency on 1k tokens after prefilling a given context length. $w=0$ denotes the model without RetroAttention. OOM denotes out-of-memory.

| Context length | Retrospective window | Throughput (batch/sec) at batch $B$ | | | | |
|---|---|---|---|---|---|---|
| | | $B$=1 | $B$=2 | $B$=4 | $B$=8 | $B$=16 |
| 32k | $w$=0 | 48.7 | 92.0 | 170.9 | 298.6 | OOM |
| | $w$=2 | 47.1 | 90.0 | 166.1 | 288.2 | OOM |
| | $w$=4 | 46.8 | 88.9 | 163.2 | 285.1 | OOM |
| | $w$=8 | 46.2 | 87.2 | 161.8 | 276.8 | OOM |
| 64k | $w$=0 | 36.6 | 68.7 | 123.6 | OOM | OOM |
| | $w$=2 | 35.6 | 67.2 | 120.5 | OOM | OOM |
| | $w$=4 | 35.4 | 66.5 | 118.3 | OOM | OOM |
| | $w$=8 | 35.1 | 65.6 | 117.2 | OOM | OOM |
| 128k | $w$=0 | 24.6 | 45.5 | OOM | OOM | OOM |
| | $w$=2 | 24.2 | 44.7 | OOM | OOM | OOM |
| | $w$=4 | 24.1 | 44.2 | OOM | OOM | OOM |
| | $w$=8 | 23.9 | 43.5 | OOM | OOM | OOM |

focus on long-generation scenarios where the total context length exceeds 32k. RetroAttention introduces a small but measurable throughput reduction. For example, at a 32k context length, throughput at batch size 1 ($B$=1) decreases by 3.3% (48.7 → 47.1 batch/sec for $w$=2). At batch size 8 ($B$=8), the reduction is 3.5% (298.6 → 288.2 batch/sec for $w$=2). The worst case occurs at $B$=8 and $w$=8, where throughput decreases by 7.3% (298.6 → 276.8 batch/sec). This is because larger retrospective windows require generating and updating more KV elements, thus increasing linear layer operations (as illustrated in Figure 4). Even in this worst case, however, the added latency is only 2.1ms/token, which is still insignificant in practice.

Longer-context settings exhibit even smaller slowdowns under their maximum feasible batch sizes. At 64k context length ($B$=4), throughput decreases by 5.2% (123.6→117.2 for $w$=8), adding 1.8ms/token. At 128k ($B$=2), the reduction is 4.4% (45.5→43.5 for $w$=8), adding 2.0ms/token. These observations further confirm that the window-dependent overhead remains limited even when $w$ is large. Finally, we emphasize that query-aware KV cache load methods, including Quest and ours, assume the full KV cache fits in GPU DRAM. In long-generation settings, this constraint naturally limits the practical batch size (typically <16 for 8B model on 80GB GPU) to avoid the KV memory explosion. Under these realistic batch-size ranges, decoding remains strongly memory-bound, allowing RetroAttention to be integrated with only marginal throughput reduction.

## E   MEMORY FOOTPRINT ANALYSIS

Memory footprint is another critical factor to consider—especially since KV cache occupancy dominates attention memory usage and constrains inference concurrency in scenarios such as offloading (Fu, 2024). It is therefore important to analyze the peak memory consumption during decoding, which typically arises from the attention operation due to the large KV cache it requires. Following the same notation in Section 2.5 (*e.g.*, the retrospective window size $w$, number of loaded KV pages $k_{\text{page}}$, Query heads $h_q$, tokens per page $P$, and head dimension $d$), the memory required for the caches involved in sparse attention computation is

$$2\, k_{\text{page}}\, P\, h_k\, d\, B \quad \text{bytes} \tag{11}$$

RetroAttention adds only small memory overhead by storing the attention outputs from the previous $w-1$ steps:

$$(w-1)\, h_q\, (d+2)\, B \tag{12}$$

bytes, including auxiliary FP32 log-sum-exp values. This term is independent of both $k_{\text{page}}$ and sequence length $L$, making its growth negligible relative to the generation length. In addition, the

Table 7: LONGGENBENCH Prompt Template

| Prompt Template |
|---|
| Answer each question step by step, adhering to the format shown in the examples provided. Start each response with 'Answer' and introduce the final response with 'The answer is'. Do not repeat the question. Ensure that you respond to all the questions presented, regardless of their number.
**Examples:**
{CoT Question_1} ⋯ {CoT Question_k}
{CoT Answer_1} ⋯ {CoT Answer_k}

**Following Question:**
{Real Question_1} ⋯ {Real Question_n} |

Table 8: **Data statistics in LONGGENBENCH**. Input and Output Tokens denote the average number of input tokens and output (generated from full-cache model) tokens.

| Benchmark | $n$-question | Input Tokens | Output Tokens |
|---|---|---|---|
| CSQA | 15 | 1337.9 | 657.8 |
| | 30 | 2029.8 | 1327.3 |
| | 45 | 2672.3 | 1964.6 |
| GSM8K | 15 | 1705.8 | 1495.0 |
| | 30 | 2649.7 | 2956.8 |
| | 45 | 3530.6 | 4238.1 |
| MMLU | 15 | 2615.7 | 951.6 |
| | 30 | 4200.8 | 1637.2 |
| | 45 | 5744.3 | 2427.5 |

masking mechanism, which records the most recent decoding step for each KV page per head, requires

$$h_k \left( \frac{L}{P} \right) \cdot 2B \tag{13}$$

bytes (assuming 32-bit integers). Overall, these additions incur minimal memory overhead compared to the KV cache and do not affect scalability.

## F  DATASET AND EXPERIMENT DETAILS

### F.1  DATASET

We primarily evaluate on LONGGENBENCH, which constructs a single input prompt by concatenating multiple reasoning questions in sequence. Table 7 displays a prompt template used for LONGGENBENCH evaluation. We follow the original implementation for the number of in-context examples ($k$) (Liu et al., 2024c). Note that this differs with the number of concatenated examples in a prompt ($n$). Also, Table 8 reports the input and output token statistics for the three datasets in LONGGENBENCH: CSQA, GSM8K, and MMLU. The number of input and output tokens grows almost linearly with the number of questions in a prompt. Reasoning-intensive benchmarks, AIME 2024, GPQA-DIAMOND, and LIVECODEBENCH-V5, are evaluated using nucleus sampling (temperature 0.6; top-$p$ 0.95) with 8, 2, and 1 samples per question, respectively. Language modeling benchmark, PG-19 is evaluated on 91 out of 100 test samples exceeding 16k tokens, using the first 16k tokens of each.

### F.2  LATENCY EVALUATION

We evaluate the latency per decoding in end-to-end and kernel breakdown manner at several context lengths. For experimental details, we set the batch size to 8 and relative budget to 0.15, and use a

---

**Algorithm 2:** UPDATEANDBUILDMASK (Single Head)

---

**Input:** page index list $S_t$; cache $\mathcal{A}$; last page index $p_{\text{curr}}$
**Output:** attention mask $M$; updated cache $\mathcal{A}$

1  $\mathcal{A}[p_{\text{curr}}] \leftarrow t-1$ ;      // Prevent current page from being attended by previous queries
2  $M \leftarrow []$;
3  **foreach** $p \in S_t$ **do**
4     |  $M$.append($\mathcal{A}[p]$);
5     |  $\mathcal{A}[p] \leftarrow t$;
6  **end**
7  **return** $M, \mathcal{A}$

---

Table 9: **Comparison with other method (ArkVale).** Accuracy results on CSQA, MMLU, and GSM8K in LONGGENBENCH with $n$=30 (*i.e.*, the number of questions in a prompt). Other experiment settings are identical with Table 1.

| Method / Benchmark | CSQA | MMLU | GSM8K |
|---|---|---|---|
| Full Attention | 74.1 | 58.7 | 60.8 |
| ArkVale | 66.2 | 57.3 | 56.2 |
| RetroAttention ($w$=2) | 69.0 | 58.2 | 61.7 |
| RetroAttention ($w$=4) | 72.2 | 57.8 | 57.6 |
| RetroAttention ($w$=8) | 71.7 | 58.2 | 58.4 |

NVIDIA A100 80G GPU. Our kernel implementation is based on FlashInfer (Ye et al., 2025), extensively optimizing CPU-side overheads such as scheduling, synchronization, and memory allocation.

Figures 6(b)–(d) present the kernel-level latency breakdown. The segments labeled *QKV* and *output project* denote the linear projections from input hidden states to Queries, Keys, and Values, and from the attention outputs to the final projected representations, respectively. $\text{score}(Q)$ corresponds to the criticality estimation step (Tang et al., 2024), where each page is scored using Quest's page digests. The *top-$k$* kernel indicates the selection of the highest-scoring $k$ pages, while *mask* represents the construction and update of the attention mask, as detailed in Algorithm 2. The terms $O_{\text{org}}$ and $O_{\text{sup}}/O_{\text{up}}$ account for two separate kernel executions: the attention output computation and the subsequent merging of cached and supplementary outputs when required. Here, $O_{\text{org}}$ denotes the attention operation time without RetroAttention, while the additional cost introduced by RetroAttention is given by the difference between the combined kernel time and $O_{\text{org}}$, denoted as $O_{\text{sup}}/O_{\text{up}}$. This term reflects the net overhead incurred in attention output computation. Finally, *KV update* reflects the cost of appending and maintaining the key–value caches.

### F.3 RETROATTENTION WITH OTHER METHOD

We further verify RetroAttention in ArkVale (Chen et al., 2024a), which is a more recent dynamic KV cache compression technique. In the experiments on LONGGENBENCH (like Table 1), RetroAttention consistently enhances the performance, highlighting its generality over other method. For example, in ArkVale, 66.2%→69.0% (without→with RetroAttention; $w$=2) on CSQA, 57.3%→58.2% on MMLU, and 56.2%→61.7% on GSM8K. The results are summarized in Table 9.

## G   RESULTS ON LONG-CONTEXT INPUT BENCHMARKS

Although our method primarily targets mitigating cumulative approximation errors during long generations, we additionally evaluated its performance under the long-context scenario commonly used in prior works. Table 10 reports task accuracy results on LONGBENCH, where we observe that the improvements over the Quest baseline are limited.

To further examine how closely sparse attention can replicate its full-attention counterpart—which is the fundamental expectation of sparse approximations—we additionally measure token-level nor-

malized Levenshtein edit similarity between sparse- and full-attention generations (Table 11). This metric explicitly quantifies the fidelity of sparse attention outputs to the gold standard full-attention version. As seen, RetroAttention consistently achieves slightly higher similarity than QUEST, but the gains remain marginal[2]. This outcome can be explained by the characteristics of the benchmark: as shown in Table 12, most LONGBENCH tasks have extremely long inputs but relatively short outputs, whereas LONGGENBENCH (Table 8) contains substantially longer generations.

Since all input tokens are processed with full attention during the prefill phase, their KV caches are exact, and approximation errors only arise during autoregressive decoding. That is, long-context input tasks exhibit far less error accumulation, because the model processes a fixed context rather than repeatedly generating and conditioning on its own imperfect outputs. As a result, retrieval accuracy becomes substantially more critical in long-context output scenarios. Consequently, when the output length is short, the cumulative error remains limited and the corrective effect of RetroAttention is less pronounced, while tasks with longer generations provide more room for improvement.

---

[2]Cases exist where accuracy drops while Levenshtein similarity rises, because the metric captures alignment with full-attention outputs that may themselves be incorrect with respect to the ground truth.

Table 10: **Comparison of Accuracy in LONGBENCH tasks.** All methods use relative KV cache budget with the minimum budget of 256 tokens.

| Method | Budget | Single-Doc QA | | | Multi-Doc QA | | | Summarization | | | Code | | Few-shot | Avg. |
|---|---|---|---|---|---|---|---|---|---|---|---|---|---|---|
| | | MF-en | NrtvQA | Qasper | 2WikiMQA | HotpotQA | Musique | GovReport | MultiNews | QMSum | LCC | RB-P | TriviaQA | |
| Full Attention | – | 54.85 | 30.11 | 45.40 | 46.40 | 55.64 | 31.28 | 35.17 | 27.15 | 25.26 | 65.12 | 58.97 | 91.65 | 47.25 |
| Quest | 0.05 | 54.53 | 30.29 | 42.54 | 45.81 | 54.40 | 31.03 | 33.81 | 26.26 | 25.10 | 62.97 | 57.16 | 91.68 | 46.30 |
| | 0.10 | 55.38 | 30.03 | 44.39 | 46.44 | 55.42 | 31.24 | 34.15 | 26.37 | 25.64 | 64.16 | 58.23 | 91.38 | 46.90 |
| | 0.15 | 54.27 | 29.99 | 45.35 | 46.29 | 55.50 | 31.27 | 34.54 | 26.64 | 25.36 | 64.06 | 58.88 | 91.49 | 46.97 |
| RetroAttention ($w$=2) | 0.05 | 53.89 | 29.11 | 43.52 | 44.57 | 54.44 | 31.46 | 33.43 | 26.20 | 25.37 | 63.65 | 57.87 | 91.66 | 46.26 |
| | 0.10 | 55.06 | 30.07 | 44.43 | 46.00 | 55.64 | 31.20 | 33.88 | 26.31 | 25.62 | 63.65 | 58.63 | 91.49 | 46.83 |
| | 0.15 | 54.56 | 29.98 | 44.47 | 46.24 | 55.39 | 31.21 | 34.42 | 26.32 | 25.36 | 63.94 | 58.98 | 91.50 | 46.86 |
| RetroAttention ($w$=4) | 0.05 | 53.08 | 29.68 | 43.67 | 44.93 | 54.42 | 30.95 | 33.29 | 26.49 | 25.83 | 63.58 | 57.43 | 91.68 | 46.25 |
| | 0.10 | 54.83 | 30.10 | 44.85 | 45.92 | 55.50 | 31.23 | 34.00 | 26.90 | 25.45 | 63.85 | 58.60 | 91.49 | 46.89 |
| | 0.15 | 54.76 | 30.10 | 44.73 | 46.19 | 55.39 | 31.20 | 34.35 | 26.72 | 25.34 | 63.72 | 59.35 | 91.50 | 46.95 |
| RetroAttention ($w$=8) | 0.05 | 53.40 | 29.74 | 42.96 | 44.90 | 54.33 | 31.29 | 33.30 | 26.54 | 25.52 | 63.47 | 57.06 | 91.68 | 46.18 |
| | 0.10 | 54.97 | 29.91 | 44.12 | 45.92 | 55.22 | 31.22 | 33.81 | 26.43 | 25.71 | 63.66 | 58.64 | 91.49 | 46.76 |
| | 0.15 | 54.78 | 30.08 | 45.06 | 46.15 | 55.39 | 31.20 | 34.40 | 26.77 | 25.63 | 63.78 | 59.21 | 91.50 | 46.99 |

Table 11: **Comparison of Normalized Levenshtein Edit Similarity in LONGBENCH tasks.** Each result is compared against its corresponding full-attention generation, and the similarity is computed at the token level. A score of 1 indicates identical outputs, and higher scores are better.

| Method | Budget | Single-Doc QA | | | Multi-Doc QA | | | Summarization | | | Code | | Few-shot | Avg. |
|---|---|---|---|---|---|---|---|---|---|---|---|---|---|---|
| | | MF-en | NrtvQA | Qasper | 2WikiMQA | HotpotQA | Musique | GovReport | MultiNews | QMSum | LCC | RB-P | TriviaQA | |
| Quest | 0.05 | 0.8920 | 0.8375 | 0.7886 | 0.9304 | 0.9582 | 0.9312 | 0.3960 | 0.3971 | 0.4843 | 0.5765 | 0.5651 | 0.5331 | 0.6908 |
| | 0.10 | 0.9187 | 0.8740 | 0.8495 | 0.9420 | 0.9721 | 0.9618 | 0.4336 | 0.4077 | 0.5387 | 0.6058 | 0.6315 | 0.6220 | 0.7298 |
| | 0.15 | 0.9149 | 0.9016 | 0.8869 | 0.9691 | 0.9749 | 0.9732 | 0.4459 | 0.4209 | 0.5692 | 0.6478 | 0.6898 | 0.6914 | 0.7571 |
| RetroAttention ($w$=2) | 0.05 | 0.8904 | 0.8470 | 0.7949 | 0.9341 | 0.9571 | 0.9291 | 0.4043 | 0.4100 | 0.4884 | 0.5904 | 0.5636 | 0.5411 | 0.6959 |
| | 0.10 | 0.9212 | 0.8748 | 0.8471 | 0.9422 | 0.9726 | 0.9651 | 0.4442 | 0.4170 | 0.5424 | 0.6127 | 0.6363 | 0.6425 | 0.7348 |
| | 0.15 | 0.9216 | 0.9008 | 0.8824 | 0.9684 | 0.9819 | 0.9744 | 0.4623 | 0.4327 | 0.5682 | 0.6430 | 0.6968 | 0.7170 | 0.7625 |
| RetroAttention ($w$=4) | 0.05 | 0.8878 | 0.8515 | 0.8108 | 0.9419 | 0.9552 | 0.9298 | 0.4076 | 0.4131 | 0.4913 | 0.5906 | 0.5664 | 0.5552 | 0.7001 |
| | 0.10 | 0.9202 | 0.8825 | 0.8574 | 0.9413 | 0.9734 | 0.9650 | 0.4401 | 0.4211 | 0.5468 | 0.6192 | 0.6337 | 0.6388 | 0.7366 |
| | 0.15 | 0.9248 | 0.9000 | 0.8811 | 0.9664 | 0.9819 | 0.9739 | 0.4665 | 0.4311 | 0.5841 | 0.6465 | 0.6903 | 0.7178 | 0.7637 |
| RetroAttention ($w$=8) | 0.05 | 0.8936 | 0.8541 | 0.7977 | 0.9386 | 0.9552 | 0.9302 | 0.4048 | 0.4161 | 0.4943 | 0.5939 | 0.5662 | 0.5531 | 0.6998 |
| | 0.10 | 0.9177 | 0.8783 | 0.8633 | 0.9433 | 0.9699 | 0.9659 | 0.4447 | 0.4217 | 0.5426 | 0.6177 | 0.6376 | 0.6486 | 0.7376 |
| | 0.15 | 0.9270 | 0.9021 | 0.8837 | 0.9677 | 0.9817 | 0.9716 | 0.4613 | 0.4266 | 0.5892 | 0.6421 | 0.6917 | 0.7139 | 0.7632 |

Table 12: **Data statistics in LONGBENCH**. Input/Output Tokens denote the average number of input tokens and output tokens (generated from the full-cache model).

| Task | Input Tokens | Output Tokens |
|---|---|---|
| MF-en | 6974.0 | 18.5 |
| NrtvQA | 29904.1 | 8.9 |
| Qasper | 5123.5 | 20.5 |
| 2WikiMQA | 7203.3 | 6.2 |
| HotpotQA | 12889.3 | 4.7 |
| Musique | 15651.5 | 6.1 |
| GovReport | 10310.4 | 459.2 |
| MultiNews | 2677.1 | 434.6 |
| QMSum | 13951.6 | 107.7 |
| LCC | 3179.7 | 64.0 |
| RB-P | 10819.4 | 64.0 |
| TriviaQA | 11780.7 | 32.0 |

