# OpenReview forum: "Retrospective Sparse Attention for Efficient Long-Context Generation"
_ICLR.cc/2026/Conference — ICLR 2026 Poster_

### Official Review · Reviewer_A5Uv · 2025-10-28

**Soundness:** 3
**Presentation:** 4
**Contribution:** 3
**Rating:** 6
**Confidence:** 5

**Summary:**

The paper “Retrospective Sparse Attention” introduces an efficient attention mechanism designed to reduce the memory and computation cost of attention without sacrificing long-range dependency modeling. Instead of attending to all past tokens, the proposed Retrospective Attention mechanism selectively attends to a subset of important historical tokens determined by sparsity patterns guided by attention statistics from previous layers or steps. This retrospective sparsity is dynamically adapted based on contextual relevance, allowing the model to focus on informative past activations while maintaining global context coverage. The method is evaluated on long-context language modeling and sequence tasks, showing comparable performance to dense attention while significantly lowering memory and compute usage. The method achieves up to 21.9% and on average 5.6% accuracy gains over state-of-the-art baselines across long-context benchmarks

**Strengths:**

1. Effective expansion of effective KV budget: Through retrospective updates, each query effectively accesses 1.6× more KV entries than the allocated cache budget, without any increase in actual memory usage
2. Clear motivation and empirical validation: The paper convincingly identifies a progressive degradation problem in long-sequence generation due to frozen past attention and directly addresses it via backward refinement
3. Novel mechanism for long-context refinement: RetroAttention effectively mitigates cumulative attention degradation during long decoding, a limitation common in prior sparse attention methods

**Weaknesses:**

1. Weak ablation and sensitivity analysis: While the method introduces retrospective window size w as a key parameter, the results (Fig. 7) show saturation beyond w=8, but there is no discussion of why or how this affects efficiency or convergence
2. Ambiguity in the update mechanism: The process of overwriting old KV entries and how it interacts with downstream layers is conceptually underexplained
3. Limited scope of benchmarks: TThere are no experiments on code-completion settings where long-context dependencies differ.

**Questions:**

1. Is the proposed method only tested on multi-questions & multi-answers? Have you tested it on multi-turn conversations? It seems to be worthwhile to apply this method on multi-turn conversations
2. How sensitive is performance to w, and what governs the optimal trade-off between accuracy and computation?

---

> ### Author Response · Authors · 2025-11-21
> **Official Comment by Authors - 1**
>
> We sincerely thank the reviewer for their insightful comments on our manuscript. We hope we have addressed their concerns and questions regarding the paper and hope they will reconsider their rating.
>
> **[W1 - Weak ablation and sensitivity analysis]. “While the method introduces retrospective window size w as a key parameter, the results (Fig. 7) show saturation beyond w=8, but there is no discussion of why or how this affects efficiency or convergence”**
>
> **Answer.** Thank you for the constructive feedback. A larger retrospective window increases the effective KV budget. Hence, the question can be translated into “why increasing the effective KV budget does not yield continuous accuracy improvements?”. This is because the most informative KV elements (those with high attention weights) are selected first. As the budget grows (i.e., as “w” increases), the additional KV elements that become included tend to have progressively lower importance. Consequently, the benefit diminishes, leading to the saturation behavior observed beyond w=8.
>
> To demonstrate this, we measured the relative attention-score gain contributed by each update. For example, in the table below (w=8 setting), the first column represents the attention scores, i.e., sum(exp(QK/sqrt(d))), at “t=0” Query (its original attention), and the next columns represent the attention scores supplemented from the next “t=1”, “t=2”, … , “t=7” Queries to this “t=0” Query. Each supplementation is normalized by its original attention scores.
>
> Empirically, the first retrospective update recovers 4–7% of the baseline attention mass, but subsequent updates contribute progressively less, indicating that later updates add only marginal signal. This is because attention weights are heavily concentrated on a small subset of tokens, expanding the retrospective window yields a non-linear and diminishing recovery of full-cache information. This directly explains the observed saturation in performance.
>
> *Table: **Relative attention-score gain by retrospective updates.** We use few sampled data in LongGenBench n-question prompts and set n=30, w=8, and budget=15%. Values in header denote the relative gain in attention mass obtained when newly arrived KV entries retrospectively update past queries (from current → past indices)*
>
> \* **t=m→n** denotes the retrospective update from m-th Query to n-th Query.
> | Benchmark | t=0→0 | t=1→0 | t=2→0 | t=3→0 | t=4→0 | t=5→0 | t=6→0 | t=7→0 |
> |-----------|-------|-------|-------|-------|-------|-------|-------|-------|
> | MMLU      | 1.000 | 0.054 | 0.018 | 0.011 | 0.007 | 0.007 | 0.005 | 0.004 |
> | GSM8K     | 1.000 | 0.044 | 0.016 | 0.009 | 0.006 | 0.004 | 0.004 | 0.004 |
> | CSQA      | 1.000 | 0.650 | 0.028 | 0.015 | 0.013 | 0.010 | 0.010 | 0.009 |
>
> In addition, the performance saturation is indeed not a specific behavior of RetroAttention. Accuracy gains by increasing the “actual” KV budget also diminish as the budget becomes larger. For example, in Quest (no RetroAttention), when the relative budget “b” is 0.10, 0.15, and 0.20, the corresponding accuracy is 40.4%, 54.9%, and 57.6% on MMLU; 32.3%, 50.9%, and 56.0% on GSM8K, as illustrated in Figure 5(c) and (d). We include this discussion on the upper-bound performance of RetroAttention in a new paragraph “Saturation in Performance Gain” in Section 3.4 and a new section in Appendix C.3. “Supplemented Attention Weights”.

---

> ### Author Response · Authors · 2025-11-21
> **Official Comment by Authors - 2**
>
> **[Q2 - Sensitive to the retrospective window, w]. “How sensitive is performance to w, and what governs the optimal trade-off between accuracy and computation?”**
>
> **Answer.** The trade-off between accuracy and computation is essentially controlled by the “w”, the only control parameter in RetroAttention. The optimal trade-off would be achieved when accuracy gain is significant (or saturated) while computation (particularly latency) overhead is still marginal.
>
> - **Accuracy**: the accuracy is most sensitive when moving from w = 0 to small windows (w = 2, 4); beyond that, gains quickly exhibit diminishing returns and largely saturate around w=8 (as also seen in Figure 5 and Figure 7). This aligns with our attention-mass analysis given in the answer above.
>
> - **Computation**: as described in Section 2.5 (Overlead Analysis), the overhead scales roughly linearly with w, as each additional update adds extra query loads and FFN activations. However, this cost remains small, as the additional computation does not significantly contribute to the latency in memory-bound regimes.
>
> This mismatch in how accuracy and overhead scale determines the practical trade-off. In typical long-generation scenarios, where decoding is strongly memory-bound, using a larger window (e.g., w=8) is generally preferred: it provides near-saturated accuracy gains while the added latency remains minimal due to the bandwidth-bound regime. However, in settings where latency is more critical or the workload is not strictly memory-bound, practitioners may choose smaller windows (w=2 or 4) to further reduce overhead while still recovering a substantial portion of the accuracy benefits. Thus RetroAttention offers a flexible knob that can be tuned depending on the runtime constraints.
>
> We include this discussion in a new section in Appendix C.4 “Sensitivity to Retrospective Window”.
>
> ---
> **[W2 -  Ambiguity in the update mechanism]. “The process of overwriting old KV entries and how it interacts with downstream layers is conceptually underexplained”**
>
> **Answer.** Please note that we have provided a separate paragraph discussing the KV cache update processes and their influences on deeper layers in Section 2.4 (L260-L271 and Figure 4). To explain details in addition to this, we incorporate the following discussion in the section:
>
> “Once the updated Key-Value (K^, V^) for the decoding steps t_1 to t_2 are overwritten in the KV cache, all subsequent decoding steps (t > t_3) automatically leverage these changes during attention. No special logic is required for deeper layers; they simply read from the KV cache as usual. This process naturally leads to a progressive reduction in attention errors across layers, as later queries attend to higher-quality representations of the past tokens.”
>
> ---
>
> **[W3 -  Limited scope of benchmarks]. “TThere are no experiments on code-completion settings where long-context dependencies differ.”**
>
> **Answer.** In fact, we have evaluated code-completion tasks using LongBench, which includes LCC (Long Code Completion) and RB-P (RepoBench-P) datasets. As mentioned in the paragraph “Long-context Input” in Section 3.4, our results are reported in Appendix G (Table 10 and Table 11). For example, the full-cache baselines exhibit 65.1% on LCC and 59.0% on RB-P. The benefits of RetroAttention are most pronounced when the relative budget “b”=0.05. Quest (b=0.5) achieves 63.0% on LCC and 57.2% on RB-P. RetroAttention with the retrospective window size “w”=2 shows 63.7% (+0.7%) on LCC and 57.9% (+0.7%) on RB-P. Interestingly, RetroAttention with b=0.15 and w=4 even surpasses the full-cache baseline, i.e., 59.4% (full cache: 59.0%). We further clarify that we have included LongBench results on single-doc and multi-doc QA, Summarization, Code, and Few-shot datasets in the paragraph “Long-context Input” in Section 3.4.

---

> ### Author Response · Authors · 2025-11-21
> **Official Comment by Authors - 3**
>
> **[Q1 - Multi-turn conversations]. “Is the proposed method only tested on multi-questions & multi-answers? Have you tested it on multi-turn conversations? It seems to be worthwhile to apply this method on multi-turn conversations”**
>
> **Answer.** As discussed in Section 3.4 (Comparison in other tasks), we have evaluated RetroAttention in other tasks as well, including reasoning tasks (AIME 2024, GPQA-Diamond, LiveCodeBench-v5), long-context input (LongBench), and language modeling (PG-19).
>
> Thank you for suggesting the experiments on multi-turn conversations. We are currently preparing these experiments, and we will include the results as soon as they become available.

---

> ### Author Response · Authors · 2025-11-28
> **Official Comment by Authors - 3 (Continue)**
>
> **[Q1 - Multi-turn conversations]. “Is the proposed method only tested on multi-questions & multi-answers? Have you tested it on multi-turn conversations? It seems to be worthwhile to apply this method on multi-turn conversations”**
>
> **Answer.** We appreciate the reviewer’s understanding regarding the delayed multi-turn experiment results.
>
> We have constructed 30-turn conversation scenarios using construct 30-turn conversations from CSQA, MMLU, and GSM8K. With a system prompt including an instruction and few-shot examples, the model alternates between taking one question as input and producing one answer as output for 30 turns. Each question input is processed with full-attention, and then its corresponding answer is generated by sparse decoding. The results are summarized in the table below.
>
> *Table: **Comparison against Quest in multi-turn conversation scenarios.** We use Llama-3.1-8B-Instruct with three relative budgets “b” of 0.15, 0.10, and 0.05 are considered and the retrospective window size "w" of 2.*
>
>
> | Method / Setting                  | CSQA   | MMLU   | GSM8K  |
> |----------------------------------|--------|--------|--------|
> | **Full-cache**                   | **66.9%** | **57.3%** | **66.7%** |
> |                                  |        |        |        |
> | Quest (b = 0.15)            | 66.4% | 57.7% | 65.0% |
> | + RetroAttention (b = 0.15)      | 67.1% | 58.1% | 65.2% |
> | Δ Retro. vs. Quest               | **+0.7%** | **+0.4%** | **+0.2%** |
> |                                  |        |        |        |
> | Quest (b = 0.10)             | 39.9% | 49.5% | 58.5% |
> | + RetroAttention (b = 0.10)      | 38.5% | 50.6% | 59.4% |
> | Δ Retro. vs. Quest               | **-1.4%** | **+1.1%** | **+0.9%** |
> |                                  |        |        |        |
> | Quest (b = 0.05)             | 30.7% | 25.6% | 9.9% |
> | + RetroAttention (b = 0.05)      | 31.9% | 27.1% | 10.8% |
> | Δ Retro. vs. Quest               | **+1.2%** | **+1.5%** | **+0.9%** |
>
>
> The results show that RetroAttention consistently matches or outperforms Quest in most settings. At the lowest budget of 0.05, RetroAttention yields the largest gain. For example, +1.2% (compared to Quest) on CSQA, +1.5% on MMLU, and 0.9% on GSM8K. As the budget increases, Quest nearly recovers full-cache performance, and the performance improvements from RetroAttention also saturates.
>
> We consider the smaller improvements than LongGenBench here are attributed to the following reasons: (1) multi-turn setting repeatedly interleaves full-attention, e.g., question (prefill: full-attention) - answer (decode: sparse attention) - question (prefill: full-attention) - ... , and (2) the model is forced to generate answers only for the current, single question. This largely "resets" the error accumulation from sparse attention unlike long-generation tasks (i.e., sparse attention continues for much longer periods).
>
> Overall, these results illustrate how RetroAttention behaves in multi-turn settings, although the method is primarily designed for long-generation tasks. We thank the reviewer again for suggesting this analysis.

---

### Official Review · Reviewer_zGZu · 2025-10-31

**Soundness:** 2
**Presentation:** 3
**Contribution:** 1
**Rating:** 2
**Confidence:** 4

**Summary:**

The paper proposes RetroAttention, a technique that retroactively updates past attention outputs using KV entries loaded at later decoding steps. The method keeps a lightweight output cache and, within a fixed retrospective window, recomputes supplementary attention for previously “unseen” KV pages, merging them with cached outputs to correct cumulative approximation errors in long generations. Claimed benefits include expanding the effective KV exposure without increasing the actual KV budget and adding only marginal latency. Experiments on LONGGENBENCH (GSM8K, MMLU, CSQA) and PG-19 are reported; the paper claims consistent gains over Quest under the same budget, and small, context-length–independent overheads.

**Strengths:**

+ The work introduces a new method for updating past KV caches using information from future tokens, allowing the model to retrospectively correct potential errors accumulated in earlier cache entries.

+ Concrete mechanism with system awareness. The design (supplementary attention + output cache + KV overwrite downstream) is explicit and connected to a memory-bound analysis; figures and pseudo-code aid clarity.

**Weaknesses:**

- Motivation not fully convincing relative to simpler practices. The paper assumes non-trivial benefits from recomputing within a small local window. However, in LServe-style serving (LServe: Efficient Long-sequence LLM Serving with Unified Sparse Attention), adjacent tokens often select the same pages—so neighboring queries tend to attend to similar KV slices. If adjacent queries already share top-k pages, retrospective supplementation within a small window may add little new signal while increasing complexity. The current empirical narrative does not isolate this effect or demonstrate cases where the new pages actually differ materially from what neighbors already saw.

- Limited breadth of models and tasks. The main results use LLAMA-3.1-8B-Instruct and swaps to DeepSeek-R1-Distill-Llama-8B for a subset of reasoning tasks. Absent are larger and diverse foundation models (eg. Qwen); conclusions about generality are therefore weak.

- Overhead and complexity vs. benefit. The method adds supplementary attention, output caching, mask bookkeeping, and KV overwrites across layers. The latency analysis argues the overhead is memory-bound and “negligible,” but even the authors’ figures indicate non-zero, window-dependent costs; moreover, these kernels complicate inference stacks and may reduce effective batch size , weakening throughput. The paper does not quantify batch-size impacts or serving-level throughput with and without the feature enabled.

- Insufficient ablations on design choices. For example, the choice of window size w, the policy for “unseen” page detection, and the exact merging schedule are only partially ablated; sensitivity to k, page size, and per-head selection policy is unclear beyond the limited plots.

- By the way, the paper states anonymized code is provided, but the link is not accessible on our side.

**Questions:**

- The core issue lies in the lack of convincing motivation. Within a block, the tokens are usually highly correlated, and in most sparse-attention implementations, the selected KV pages for neighboring tokens are largely identical. You should discuss in detail the perspective presented in the LServe paper, which shows that adjacent tokens typically attend to the same KV pages. To strengthen your argument, please provide additional experiments that clearly demonstrate the necessity of updating the KV cache within a sliding window and justify why such retrospective updates are beneficial.

- All other detailed comments can be found in the Weaknesses section.

---

> ### Author Response · Authors · 2025-11-21
> **Official Comment by Authors - 1**
>
> We sincerely thank the reviewer for their insightful comments on our manuscript. We hope we have addressed their concerns and questions regarding the paper and hope they will reconsider their rating.
>
> **[W1 - Motivation is not convincing]. “Motivation not fully convincing relative to simpler practices. The paper assumes non-trivial benefits from recomputing within a small local window. However, in LServe-style serving (LServe: Efficient Long-sequence LLM Serving with Unified Sparse Attention), adjacent tokens often select the same pages—so neighboring queries tend to attend to similar KV slices. If adjacent queries already share top-k pages, retrospective supplementation within a small window may add little new signal while increasing complexity.”**
>
> **Answer.** Thank you for the constructive feedback. Our motivation primarily arises in long-generation scenarios, where seemingly small KV cache supplementation can be repeated throughout longer decoding steps and ultimately yield non-trivial performance gains.
>
> As the reviewer mentioned, LServe-style serving (i.e., reusing previous top-K pages) may well approximate the current query’s KV cache. However, as the LServe paper (its Table 6) already exhibits some degradation even with “Reusing Interval” of 1, there often exist non-overlapped pages as well. While we agree that these amounts may not be significant, an important question is whether we can simply overlook these few non-overlapped pages. Our answer is no, especially in long-generation scenarios.
>
> Long-generation tasks are more sensitive to budget: Our key observation shows that, in LongBench (long-input & short-output), Quest with the 5% budget already exhibits comparable accuracy (i.e., 46.3%) to the full-cache baseline (i.e., 47.3%). This implies that reducing or increasing the budget leads to only marginal influences on this accuracy. In contrast, in LongGenBench (long-input & long-output), with the same 5% budget, Quest encounters significant degradation (e.g., 60.8% -> 17.6% on GSM8K). More importantly, by increasing the budget to 10%, 15%, and 20%, the accuracy becomes 32.3%, 50.9%, and 56.0%. This trend is because either missing or supplementing KV cache in long generation repeats over far more decoding steps, amplifying their influences.
>
> Such difference motivates us to consistently supplement missing KV cache, efficiently within a small local window, as they will benefit across longer decoding steps and effectively mitigates the error accumulation. We add this discussion in Section 2.2. Motivational Study.

---

> ### Author Response · Authors · 2025-11-21
> **Official Comment by Authors - 2**
>
> **[W1 - Motivation is not convincing]. “The current empirical narrative does not isolate this effect or demonstrate cases where the new pages actually differ materially from what neighbors already saw.”**
>
> **[Q1 - Motivation is not convincing]. “You should discuss in detail the perspective presented in the LServe paper, which shows that adjacent tokens typically attend to the same KV pages. To strengthen your argument, please provide additional experiments that clearly demonstrate the necessity of updating the KV cache within a sliding window and justify why such retrospective updates are beneficial.”**
>
> **Answer.** Continuing the previous answer, we would like to strengthen our argument through the following experimental evidence: (1) 20% of top-k pages are not shared in adjacent queries, and (2) LServe-style approach is indeed vulnerable in long generation. We add a separate section in Appendix (B. Motivational Study) to discuss them.
>
> **(1) How many top-K pages are shared in adjacent queries?**: We demonstrate that adjacent queries share 82.8% (interval: 1) to 73.8% (interval: 8) top-K pages in Figure 2(b). In other words, more than a quarter of top-K pages “do not overlap” in the 8-step interval, which we do not consider little, particularly given that we can leverage them without increasing the actual KV budget. In addition, Figure 2(c) describes that retrospective supplementation is influenced by all the past queries in the window (not like a single query influences only one), reaching 17.2% (window: 1) to 60% (window: 8) increases in the effective KV budget. We confirm such trends hold across multiple datasets, including GSM8K, CSQA, and MMLU (see Figure 8 in Appendix).
>
> **(2) Does LServe-style method perform well in long generation?**: We believe that the reviewer’s concern appears to be LServe-style methods (i.e., reusing the previous top-K pages) empirically perform well, assuming missing few non-overlapping pages (according to (1), about 20% top-K pages) by the reusage would not affect the performance much. However, their evaluation has largely been on long-input scenarios (e.g., LongBench, NIAH). To refute this assumption in long-generation tasks, we apply the LServe style (i.e., reusing top-k pages) to Quest in LongGenBench. We observe that LServe-style Quest significantly degrades the accuracy, even at the smallest reuse frequency of 2. For example, Quest without and with the reuse exhibit 52.6% and 25.2% (average accuracy) on GSM8K, 54.8% and 34.7% on MMLU, and 56.8% and 29.2% on CSQA (see table below). This highlights that even if the amount of missed or supplemented KV cache is seemingly small, their influences can be large in long-generation scenarios. We include these results in Table 4 (Appendix).
>
> *Table: **Comparison against Top-K reusage**. “LServe-style” reuses the previously loaded top-k pages and is integrated into Quest. Here, f=2 indicates that top-k pages are refreshed every two
> decoding steps. We set the relative budget b to 0.15.*
>
> \* The number next to the dataset name (e.g., 15, 30, and 45) means the number of questions in a prompt.
> | Method                    | *GSM8K* 15 | 30  | 45  | Mean | *MMLU* 15 | 30  | 45  | Mean | *CSQA* 15 | 30  | 45  | Mean |
> |---------------------------|------------|-----|-----|------|-----------|-----|-----|------|-----------|-----|-----|------|
> | Full-cache                | 66.7       | 60.8 | 58.0 | 61.8 | 62.5      | 58.7 | 57.1 | 59.4 | 73.5      | 74.1 | 71.9 | 73.2 |
> | Quest                     | 58.2       | 50.9 | 48.6 | 52.6 | 58.8      | 54.9 | 50.6 | 54.8 | 70.9      | 53.5 | 46.1 | 56.8 |
> | + LServe-style (f=2)      | 22.4       | 22.3 | 31.0 | 25.2 | 39.6      | 32.5 | 32.0 | 34.7 | 40.7      | 25.1 | 21.9 | 29.2 |
> | + RetroAttention (w=2)    | 61.3       | 52.6 | 55.4 | 56.5 | 59.3      | 55.4 | 51.2 | 55.3 | 68.8      | 59.1 | 53.0 | 60.3 |

---

> ### Author Response · Authors · 2025-11-21
> **Official Comment by Authors - 3**
>
> **[W2 - Limited breadth of models and tasks]. “Absent are larger and diverse foundation models (eg. Qwen); conclusions about generality are therefore weak.”**
>
> **Answer.** We further validate RetroAttention in two additional models: Qwen2.5-14B-Instruct and Qwen2.5-32B-Instruct. We evaluate their performance with and without RetroAttention using LongGenBench, following the same experimental settings in Table 1 (# questions “n”: 15, 30, and 45; relative budget “b”: 0.15, retrospective window size “w”: 2). RetroAttention still exhibits consistent accuracy improvements across the three datasets (GSM8K, MMLU, and CSQA). In the 14B model, 58.1% (no RetroAttention) -> 65.5% (RetroAttention; +7.4%) on CSQA, 73.2% -> 77.6% (+4.4%) on GSM8K, and 73.3% -> 73.5% (+0.2%) on MMLU. In the 32B model, 75.4% -> 79.7% (+4.3%) on CSQA, 83.8% -> 87.8% (+4.0%) on GSM8K, and 80.0% -> 80.4% (+0.4%) on MMLU. These results support the RetroAttention’s generality in larger models. We include the results and discussion in a new Table 2 and new paragraph “Accuracy Improvements in Larger Models” in Section 3.2.
>
> *Table: **Comparison in larger models.** Accuracy results in Qwen2.5-32B-Instruct (top) and Qwen2.5-14B-Instruct (bottom) models. Experiments settings are identical with Table 1 (main paper).*
>
> ### Qwen2.5-32B
> \* The number next to the dataset name (e.g., 15, 30, and 45) means the number of questions in a prompt.
> | Method                | *GSM8K* 15 | 30  | 45  | Mean | *MMLU* 15 | 30  | 45  | Mean | *CSQA* 15 | 30  | 45  | Mean |
> |----------------------|--------------|-----|-----|------|-------------|-----|-----|------|-------------|-----|-----|------|
> | Full-cache           | 92.1 | 91.7 | 90.8 | 91.5 | 81.8 | 81.0 | 81.2 | 81.3 | 86.1 | 87.0 | 87.1 | 86.7 |
> | Quest                | 89.2 | 78.8 | 83.3 | 83.8 | 81.0 | 79.4 | 79.5 | 80.0 | 82.3 | 69.3 | 74.4 | 75.4 |
> | RetroAttention (w=2) | 89.8 | 89.1 | 84.4 | 87.8 | 80.9 | 80.3 | 79.9 | 80.4 | 85.3 | 75.0 | 78.7 | 79.7 |
> | Δ Retro vs. Que.     | 0.6  | 10.3 | 1.1  | 4.0  | -0.1 | 1.0  | 0.4  | 0.4  | 2.8  | 5.7  | 4.3  | 4.3  |
>
> ### Qwen2.5-14B
> \* The number next to the dataset name (e.g., 15, 30, and 45) means the number of questions in a prompt.
> | Method                | *GSM8K* 15 | 30  | 45  | Mean | *MMLU* 15 | 30  | 45  | Mean | *CSQA* 15 | 30  | 45  | Mean |
> |----------------------|--------------|-----|-----|------|-------------|-----|-----|------|-------------|-----|-----|------|
> | Full-cache           | 88.6 | 84.4 | 76.6 | 83.2 | 78.2 | 76.6 | 73.6 | 76.1 | 83.7 | 83.5 | 83.5 | 83.6 |
> | Quest                | 79.0 | 76.5 | 64.1 | 73.2 | 76.0 | 73.0 | 71.1 | 73.3 | 69.4 | 46.8 | 58.1 | 58.1 |
> | RetroAttention (w=2) | 83.1 | 80.7 | 69.1 | 77.6 | 75.8 | 73.7 | 71.0 | 73.5 | 73.3 | 59.5 | 63.6 | 65.5 |
> | Δ Retro vs. Que.     | 4.1  | 4.2  | 5.0  | 4.4  | -0.2 | 0.7  | -0.1 | 0.2  | 3.9  | 12.8 | 5.5  | 7.4  |

---

> ### Author Response · Authors · 2025-11-21
> **Official Comment by Authors - 4**
>
> **[W3 - Overhead and Complexity vs. Benefits]. “The latency analysis argues the overhead is memory-bound and “negligible,” but even the authors’ figures indicate non-zero, window-dependent costs; moreover, these kernels complicate inference stacks and may reduce effective batch size , weakening throughput. The paper does not quantify batch-size impacts or serving-level throughput with and without the feature enabled.”**
>
> **Answer.** Thanks for pointing out an important point. To answer the impact on batching size, we analyze the decoding throughput with and without RetroAttention across a range of batch sizes and long-context settings (see the table below). Because recent models (e.g., reasoning-oriented) often generate more than 10k tokens, we focus on long-generation scenarios where the total context length exceeds 32k. We include the following discussion and results in a new paragraph “Batch-size impacts on decoding throughput” (Appendix D) and Table 6 (Appendix).
>
> As anticipated by the reviewer, RetroAttention introduces a small but measurable throughput reduction. For example, at a 32k context length, throughput at batch size 1 (B=1) decreases by 3.3% (48.7 -> 47.1 batch/sec for w=2). At batch size 8 (B=8), the reduction is 3.5% (298.6 -> 288.2 batch/sec for w=2). The worst case occurs at B=8 and w=8, where throughput decreases by 7.3% (298.6 -> 276.8 batch/sec). This is because larger retrospective windows require generating and updating more KV elements, thus increasing linear layer operations (as illustrated in Figure 4). Even in this worst case, however, the added latency is only 2.1ms/token, which is still insignificant in practice.
>
> Longer-context settings exhibit even smaller slowdowns under their maximum feasible batch sizes. At 64k context length (B=4), throughput decreases by 5.2% (123.6 -> 117.2 for w=8), adding 1.8ms/token. At 128k (B=2), the reduction is 4.4% (45.5 -> 43.5 for w=8), adding 2.0ms/token. These observations further confirm that the window-dependent overhead remains limited even when w is large.
>
> Finally, we emphasize that query-aware KV cache load methods, including Quest and ours, assume the full KV cache fits in GPU DRAM. In long-generation settings, this constraint naturally limits the practical batch size (typically <16 for 8B model on 80GB GPU) to avoid the KV memory explosion. Under these realistic batch-size ranges, decoding remains strongly memory-bound, allowing RetroAttention to be integrated with only marginal throughput reduction.
>
> *Table: **batch-size impacts on throughput(batch/second) across context lengths and retrospective windows.** Experiments are performed in Llama3.1-8B-Instruct model using a single A100 80GB GPU. Throughput is estimated by estimating the average decoding latency on 1k tokens after prefilling a given context length. “w=0” denotes the model without RetroAttention.*
>
> \* The numbers denote **Throughput (batch / second) at batch B**.
> | context length | retrospective window | B=1 | B=2 | B=4 | B=8 | B=16 |
> |----------------|-----------------------|-----|-----|-----|-----|------|
> | **32k**        | w=0 | 48.7 | 92.0 | 170.9 | 298.6 | OOM |
> |                | w=2 | 47.1 | 90.0 | 166.1 | 288.2 | OOM |
> |                | w=4 | 46.8 | 88.9 | 163.2 | 285.1 | OOM |
> |                | w=8 | 46.2 | 87.2 | 161.8 | 276.8 | OOM |
> | **64k**        | w=0 | 36.6 | 68.7 | 123.6 | OOM | OOM |
> |                | w=2 | 35.6 | 67.2 | 120.5 | OOM | OOM |
> |                | w=4 | 35.4 | 66.5 | 118.3 | OOM | OOM |
> |                | w=8 | 35.1 | 65.6 | 117.2 | OOM | OOM |
> | **128k**       | w=0 | 24.6 | 45.5 | OOM | OOM | OOM |
> |                | w=2 | 24.2 | 44.7 | OOM | OOM | OOM |
> |                | w=4 | 24.1 | 44.2 | OOM | OOM | OOM |
> |                | w=8 | 23.9 | 43.5 | OOM | OOM | OOM |

---

> ### Author Response · Authors · 2025-11-21
> **Official Comment by Authors - 5**
>
> **[W4 - Insufficient ablations on design choices]. “For example, the choice of window size w, the policy for “unseen” page detection, and the exact merging schedule are only partially ablated; sensitivity to k, page size, and per-head selection policy is unclear beyond the limited plots.”**
>
> **Answer.** For the policy for “unseen” page detection, and the exact merging schedule, we apologize for the confusion. Discussions on the policy for “unseen” page detection have been included in Appendix C.2 (“Retrospective Mask” paragraph). As mentioned in L231-L232, the merging of original and supplementary attention outputs are detailed in Appendix C (particularly, Equation (6)-(9)), and their schedule is described in Algorithm 1 in Appendix C.1 (“RetroAttention Algorithm Flow”). We now clearly refer to these details from Section 2.3 (L234-L236) of the main paper.
>
> Please note that, for the choice of window size “w”, we have provided the ablations on the window size w=2, 4, 8 across three different KV cache budgets (b=0.10, 0.15, 0.20) and three different benchmarks (CSQA, MMLU, and GSM8K) in Figure 5. In addition, corresponding latency analysis on each window size w=2, 4, 8 is presented in Figure 6. Furthermore, Figure 7(b) confirms the performance gain saturates at w=8 and 16.
>
> While sensitivity to k and page size would be important control parameters, these are more closely related to the ablations of the original “Quest” method, which is beyond the scope of out study. For per-head selection, since Quest method is limited to MHA not GQA, we made it applicable by selecting pages KV-head-wise, by aggregating (averaging) Quest scores across group sizes. Page size of 16 is chosen since it is the default setting of Quest.
>
> ---
>
> **[W5 - Code link]. “By the way, the paper states anonymized code is provided, but the link is not accessible on our side.”**
>
> **Answer.** We sincerely apologize for the issues with the original link. Although we confirmed that it was working well at the time of submission, it appears that unexpected errors occurred afterward. We are providing a new anonymized code link containing the exact same contents as submitted: https://anonymous.4open.science/r/RetroAttention-1940. The link in Appendix A (Overview) is also updated. Thank you for your understanding.

---

### Official Review · Reviewer_ytzA · 2025-11-01

**Soundness:** 3
**Presentation:** 3
**Contribution:** 4
**Rating:** 8
**Confidence:** 2

**Summary:**

RetroAttention targets the long-context inference bottleneck, the fact that the KV cache that grows linearly with sequence length, and the cumulative errors introduced by sparse/compressed attention during extended decoding. The method retrospectively revises past attention outputs using KV entries fetched at later decoding steps. It does this by computing the supplementary attention for earlier queries with currently loaded but previously unseen KV pages, and storing/updating these results in a lightweight output cache. Original and supplementary outputs are merged via a softmax-weighted linear combination, and the refined embeddings propagate upward by overwriting stale K/V entries in deeper layers.

**Strengths:**

1. RetroAttention has consistent gains over strong baselines while adding only marginal latency thanks to its memory-bound design.

2. The scheme expands each query's effective KV budget by up to 1.6x without increasing the actual KV budget or KV memory traffic. This helps improve contextual completeness under fixed memory. This is a very nice approach and I liked this part.

**Weaknesses:**

1. This method works only long-generation scenarios. This approach wont help if KV cache overheads are not a problem (such as cases using extreme quantization).

2. An interesting fact is that the methods are not very simple (which would have been preferred). The benefits saturate as the retrospective window increases (nearly w=8). The approach introduces extra kernel steps (mask management, output merges, KV overwrites) that, while small, add complexity and non-zero overhead.

**Questions:**

1. Could RetroAttention be combined with eviction-style key selection (e.g., Keyformer) or MorphKV (Dialogue without limits paper) so that retrospective updates counteract the cumulative errors introduced by permanently discarded tokens, rather than conflicting with eviction's objectives?

2. Given the limited gains on long-input/short-output settings like LONGBENCH, would token-selection methods (Keyformer/MorphKV or similar strategies) fare better there? Also, could a hybrid (selection + retrospective updates) recover benefits across both regimes?

---

> ### Author Response · Authors · 2025-11-21
> **Official Comment by Authors - 1**
>
> We sincerely thank the reviewer for their insightful comments on our manuscript. We hope we have addressed their concerns and questions regarding the paper and hope they will reconsider their rating.
>
> **[W1 - Works only for long-generation scenarios?]. “This method works only long-generation scenarios. This approach wont help if KV cache overheads are not a problem (such as cases using extreme quantization).”**
>
> **Answer.** RetroAttention works in any tasks (as the method itself is task-agnostic), but we agree with the reviewer that its latency gain would be less effective if KV cache overheads are reduced. However, this is true in any other KV cache compression techniques, and hence we do not consider this as a specific weakness of RetroAttention. In addition, our approach is orthogonal to KV quantization, and thus, RetroAttention can be combined with KV quantization without conflicts. The two methods address different sources of overhead (total number of tokens to load vs. per-token cost to load) and can provide cumulative benefits.
>
> ---
>
> **[W2 - Methods are not very simple]. “An interesting fact is that the methods are not very simple (which would have been preferred). The benefits saturate as the retrospective window increases (nearly w=8). The approach introduces extra kernel steps (mask management, output merges, KV overwrites) that, while small, add complexity and non-zero overhead.”**
>
> **Answer.** We appreciate the reviewer’s concern regarding the added complexity and saturated benefits as the retrospective window grows. Our method, however, introduces only marginal overhead. For example, the latency overhead remains very small: approximately 2ms per token for w=8, which is practically negligible. Importantly, this overhead is constant, independent of context length or the size of the retrieved cache, as analyzed in Section 2.5. In addition, even though the performance improvement saturates around w=8, the gains at this window size are already significant. For example, +16.3% on CSQA, +5.0% on GSM8K, and +1.5% on MMLU (please see Figure 5(a)).
>
> ---
>
> **[Q1 - Combination with eviction-style key selection]. “Could RetroAttention be combined with eviction-style key selection (e.g., Keyformer) or MorphKV (Dialogue without limits paper) so that retrospective updates counteract the cumulative errors introduced by permanently discarded tokens, rather than conflicting with eviction's objectives?”**
>
> **Answer.** Thank you for the interesting question. Our answer is no, RetroAttention is designed for query-aware (non-eviction) key selection techniques. As mentioned in L213, our supplementary attention output is computed on “currently loaded, but previously unseen” KV cache. However, in eviction-style key selections (both KeyFormer and MorphKV), once KV cache is evicted, these can never be loaded and attended to by the future queries. That is, all the currently loaded KV cache has been used by the previous attention, and hence, “supplementation” of KV cache cannot be established. We further clarify that RetroAttention is specifically designed for query-aware KV cache compression techniques in Introduction (L73-L76).

---

> ### Author Response · Authors · 2025-11-21
> **Official Comment by Authors - 2**
>
> **[Q2 - Comparison with eviction-style methods in LongBench]. “Given the limited gains on long-input/short-output settings like LONGBENCH, would token-selection methods (Keyformer/MorphKV or similar strategies) fare better there?”**
>
> **Answer.** We do not think token-selection methods fare better. To support this, we mainly compare MorphKV’s performance on LongBench with our method. MorphKV, with the relative budget of 0.20, shows -2.51% average degradation than the full-cache (from Table 3 in the MorphKV paper). In contrast, RetroAttention, with the relative budget of 0.15 (less than MorphKV), exhibits -0.29% average degradation than the full-cache. We observe that Quest already reaches -0.39% average degradation (see the table below). Hence, the better performance is primarily driven by the query-aware KV selection method, and RetroAttention additionally mitigates the performance degradation.
>
> We could not find the LongBench results from Keyformer. Our comparison is made in the following LongBench categories: MF-en (mfqaen), NrtvQA (nqa), Qasper (qsp), 2WikiMQA (2wmqa), HotpotQA (hpqa), Musique (musq), MultiNews (mnews), QMSum (qms), TriviaQA (tqa). Those not evaluated in both methods are excluded. In terms of accuracy, ours is better due to the superior performance of query-aware KV selection methods (even with less budget). However, eviction-based methods can reduce the KV cache memory to store.
>
> *Table: **LongBench results of Llama-3.1-8B-Instruct from the MorphKV paper:** budget=0.20, degradation=-2.5% (compared to full-cache)*
>
> | Method    | mfqaen | nqa  | qsp  | 2wmqa | hpqa | musq | mnews | qms  | tqa  | Average |
> |-----------|--------|------|------|-------|------|------|-------|------|------|---------|
> | Full-cache | 27.4  | 32.0 | 13.2 | 16.5  | 16.7 | 11.4 | 26.8  | 23.6 | 91.6 | 28.80   |
> | MorphKV    | 25.7  | 31.9 | 11.9 | 14.9  | 15.9 | 10.7 | 26.6  | 23.6 | 91.5 | 28.08   |
>
> *Table: **Our LongBench results:** budget=0.15, w=8, Llama-3.1-8B-Instruct, degradation=-0.29% (compared to full-cache)*
> | Method      | mfqaen | nqa  | qsp  | 2wmqa | hpqa | musq | mnews | qms  | tqa  | Average |
> |-------------|--------|------|------|-------|------|------|-------|------|------|---------|
> | Full-cache  | 54.9   | 30.1 | 45.4 | 46.4  | 55.6 | 31.3 | 27.2  | 25.3 | 91.7 | 45.32   |
> | Quest       | 54.3   | 30.0 | 45.4 | 46.3  | 55.5 | 31.3 | 26.6  | 25.4 | 91.5 | 45.14   |
> | Ours        | 54.8   | 30.1 | 45.1 | 46.2  | 55.4 | 31.2 | 26.8  | 25.6 | 91.5 | 45.19   |
>
> ---
> **[Q2 - Comparison with eviction-style methods in LongBench]. “Also, could a hybrid (selection + retrospective updates) recover benefits across both regimes?”**
>
> **Answer.** As we explained in the previous question (Q1), RetroAttention does not support the hybridization of eviction-style selection and retrospective updates.

---

> > ### Comment · Reviewer_ytzA · 2025-11-27
> >
> > Thank you for the clear, technical, and intuitive replies. I have read your responses and I am optimistic of this paper.

---

> ### Author Response · Authors · 2025-11-28
>
> Dear reviewer ytzA.
>
> We sincerely appreciate your careful feedback on our work, as well as the positive assessment of our additional responses and the updated manuscript.
>
> Sincerely,
>
> Authors

---

### Official Review · Reviewer_zHKN · 2025-11-02

**Soundness:** 3
**Presentation:** 3
**Contribution:** 3
**Rating:** 6
**Confidence:** 4

**Summary:**

This paper introduces RetroAttention, a accuracy-refinement method for efficient long-context generation in large language models (LLMs). Unlike prior approaches (e.g., Quest, StreamingLLM, TOVA), which focus solely on compressing or sparsifying the current decoding step, RetroAttention retrospectively revises past attention outputs using newly arrived Key-Value (KV) entries from later decoding steps. Experiments across multiple long-generation benchmarks (LONGGENBENCH, AIME24, GPQA-DIAMOND, LIVECODEBENCH, and PG-19) show that RetroAttention improves effective KV exposure by up to 1.6× and yields up to 21.9% accuracy gains over state-of-the-art baselines, with small latency and memory overhead.

**Strengths:**

S1. This paper tackles an important problem of the refinement of sparse KV cache selection.

S2. In stead of proposing KV cache sparsification approaches, this paper proposes a refinement approach for the KV cache sparsificaitons, which has the potential to be applied to many backbone sparsifications.

S3. The proposed approach is a backed up with formal time and space overhead analysis.

**Weaknesses:**

W1. The experiments are limited to small scale models. Experiments on at least medium size, e.g., 32B models, would be useful.

W2. The comparison approaches are limited. More recent KV cache compression approaches, like ICECache, ArkVale, PQCache, MagicPIG, etc, should also be evaluated.

W3. I would suggest to also test larger cache reuse depth, e.g., w > 5, to show the effectiveness of reusage.

**Questions:**

Q1. It is not straightforward why the proposed approach is less effective on long context, which basically also suffers from the lower retrieval accuracy problem.

Q2. Would you also consider to use the retrieved context of the previous tokens to enhance the current tokens, since they are already in the GPU memory and they are expected to be in the top-k/2k/3k window? Why or why not?

Q3. Whether RetroAttention will work on distributed decoding?

Q4. How would RetroAttention work on MoE models?

Q5. How would RetroAttention interplay with different KV cache retrieval strategies, like those listed in W3?

---

> ### Author Response · Authors · 2025-11-21
> **Official Comment by Authors - 1**
>
> We sincerely thank the reviewer for their insightful comments on our manuscript. We hope we have addressed their concerns and questions regarding the paper and hope they will reconsider their rating.
>
> **[W1 - Experiments on larger models]. “The experiments are limited to small scale models. Experiments on at least medium size, e.g., 32B models, would be useful.”**
>
> **Answer.** Thank you for the suggestion. We additionally validate RetroAttention in Qwen2.5-32B-Instruct and Qwen2.5-14B-Instruct models. Our results show that RetroAttention is still effective in the medium-size models. Following the LongGenBench-based experiment conditions in Table 1, we extensively evaluate the models with the varying number of Question-Answer pairs (“n”=15, 30, and 45). For example, Qwen2.5 32B model with RetroAttention (w=2, b=0.15) improves the Quest baseline on average by +4.3% on CSQA, +4.0% on GSM8K, and +0.4% on MMLU (see the table below). In the 14B model, the average improvements are +7.4% on CSQA, +4.4% on GSM8K, and +0.2% on MMLU. Considering the improvement over Quest in Llama3.1-8B is +3.5% on CSQA, +3.9% on GSM8K, and +0.5% on MMLU, RetroAttention is still effective in the 32B model. We add and discuss these results in Table 2 and in a new paragraph “Accuracy Improvements in Larger Models” in Section 3.2.
>
> *Table: **Comparison in larger models.** Accuracy results in Qwen2.5-32B-Instruct (top) and Qwen2.5-14B-Instruct (bottom) models. Experiments settings are identical with Table 1 (main paper).*
>
> ### Qwen2.5-32B
>
> \* The number next to the dataset name (e.g., 15, 30, and 45) means the number of questions in a prompt.
> | Method                | *GSM8K* 15 | 30  | 45  | Mean | *MMLU* 15 | 30  | 45  | Mean | *CSQA* 15 | 30  | 45  | Mean |
> |----------------------|--------------|-----|-----|------|-------------|-----|-----|------|-------------|-----|-----|------|
> | Full-cache           | 92.1 | 91.7 | 90.8 | 91.5 | 81.8 | 81.0 | 81.2 | 81.3 | 86.1 | 87.0 | 87.1 | 86.7 |
> | Quest                | 89.2 | 78.8 | 83.3 | 83.8 | 81.0 | 79.4 | 79.5 | 80.0 | 82.3 | 69.3 | 74.4 | 75.4 |
> | RetroAttention (w=2) | 89.8 | 89.1 | 84.4 | 87.8 | 80.9 | 80.3 | 79.9 | 80.4 | 85.3 | 75.0 | 78.7 | 79.7 |
> | Δ Retro vs. Que.     | 0.6  | 10.3 | 1.1  | 4.0  | -0.1 | 1.0  | 0.4  | 0.4  | 2.8  | 5.7  | 4.3  | 4.3  |
>
> ### Qwen2.5-14B
>
> \* The number next to the dataset name (e.g., 15, 30, and 45) means the number of questions in a prompt.
> | Method                | *GSM8K* 15 | 30  | 45  | Mean | *MMLU* 15 | 30  | 45  | Mean | *CSQA* 15 | 30  | 45  | Mean |
> |----------------------|--------------|-----|-----|------|-------------|-----|-----|------|-------------|-----|-----|------|
> | Full-cache           | 88.6 | 84.4 | 76.6 | 83.2 | 78.2 | 76.6 | 73.6 | 76.1 | 83.7 | 83.5 | 83.5 | 83.6 |
> | Quest                | 79.0 | 76.5 | 64.1 | 73.2 | 76.0 | 73.0 | 71.1 | 73.3 | 69.4 | 46.8 | 58.1 | 58.1 |
> | RetroAttention (w=2) | 83.1 | 80.7 | 69.1 | 77.6 | 75.8 | 73.7 | 71.0 | 73.5 | 73.3 | 59.5 | 63.6 | 65.5 |
> | Δ Retro vs. Que.     | 4.1  | 4.2  | 5.0  | 4.4  | -0.2 | 0.7  | -0.1 | 0.2  | 3.9  | 12.8 | 5.5  | 7.4  |

---

> ### Author Response · Authors · 2025-11-21
> **Official Comment by Authors - 2**
>
> **[W2 - Comparison approaches are limited]. “More recent KV cache compression approaches, like ICECache, ArkVale, PQCache, MagicPIG, etc, should also be evaluated.”**
>
> **[Q5 - RetroAttention with different KV cache retrieval methods]. “How would RetroAttention interplay with different KV cache retrieval strategies, like those listed in W3?”**
>
> **Answer.** As the reviewer suggested, we further verify RetroAttention in ArkVale [1], which is a more recent dynamic KV cache compression technique. In LongGenBench-based experiments (like Table 1), our RetroAttention consistently enhances the performance, highlighting its generality over various methods. For example, in ArkVale, 66.2% -> 69.0% (without -> with RetroAttention; w=2) on CSQA, 57.3% -> 58.2% on MMLU, and 56.2% -> 61.7% on GSM8K (see the table below). We include these results in Appendix F.3 “RetroAttention with other method” and Table 9 (Comparison with other method (ArkVale)).
>
> *Table: **Comparison with other method (ArkVale).** Accuracy results on CSQA, MMLU, and GSM8K in LongGenBench with n = 30 (i.e., the number of questions in a prompt). Other experiment settings are identical with Table 1 (main paper).*
>
> ### Llama-3.1-8B-Instruct
>
> | Method                       | CSQA | MMLU | GSM8K |
> |------------------------------|------|------|-------|
> | Full-cache                   | 74.1 | 58.7 | 60.8  |
> | ArkVale (budget=0.15)        | 66.2 | 57.3 | 56.2  |
> | + RetroAttention (w=2)       | 69.0 | 58.2 | 61.7  |
> | + RetroAttention (w=4)       | 72.2 | 57.8 | 57.6  |
> | + RetroAttention (w=8)       | 71.7 | 58.2 | 58.4  |
>
>
> ----
> [1] Arkvale: Efficient Generative LLM Inference with Recallable Key-Value Eviction.", NeurIPS 2024.

---

> ### Author Response · Authors · 2025-11-21
> **Official Comment by Authors - 3**
>
> **[W3 - Larger retrospective window size]. “I would suggest to also test larger cache reuse depth, e.g., w > 5, to show the effectiveness of reusage.”**
>
> **Answer.** We have already tested RetroAttention with larger window sizes (w > 5), as mentioned in L413-L416 (w=8), Figure 5 (w=8), and Figure 7 (w=8 and 16). RetroAttention further reduces the performance gap between the full-cache baseline and the compressed model as w increases (as discussed in L413-L416), but at the same time, the improvement saturates beyond w > 8 (as discussed in L484-485).
>
> ---
>
> **[Q1 - Why less effective on long-context]. “It is not straightforward why the proposed approach is less effective on long context, which basically also suffers from the lower retrieval accuracy problem.”**
>
> **Answer.** We agree with the reviewer that both long-context “input” tasks (e.g., LongBench) and long-context “output” tasks (e.g., LongGenBench) are influenced by retrieval accuracy. However, as noted in Appendix G (L1194-L1195, L1199-L1200), errors from incorrect retrieval (i.e., low-quality decoding) can repeat and accumulate over much more generation steps in long-context output settings. In contrast, long-context input tasks exhibit far less error accumulation, because the model processes a fixed context rather than repeatedly generating and conditioning on its own imperfect outputs.
>
> As a result, retrieval accuracy becomes substantially more critical in long-context output scenarios. RetroAttention is specifically designed to mitigate these cumulative errors, which explains why the improvements are more pronounced in long-context output tasks and appear less prominent in long-context input tasks. We strengthen the discussion on the differences between the two tasks in the Appendix G (Results on Long-Context Input Benchmarks).
>
> ---
>
> **[Q2 - Updates on “current” tokens from previous ones]. “Would you also consider to use the retrieved context of the previous tokens to enhance the current tokens, since they are already in the GPU memory and they are expected to be in the top-k/2k/3k window? Why or why not?”**
>
> **Answer.** Thank you for the interesting question. In a nut shell, we do not consider using the previous queries’ retrieved context to enhance the current one, since such implementation will proportionally increase the memory traffic.
>
> Let’s consider a scenario where a query at t0 computes its attention output using the KV cache of KV_0, KV_1, and KV_2, and then another query at t1 uses KV_0, KV_1, and KV_3 for its attention. The reviewer’s question would be “Can we use KV_2 loaded by the query at t0 to enhance the attention output of the query at t1?”.
>
> This necessitates that the supplementary attention output (O_sup) solely on KV_2 to be pre-computed “at t0” so that the attention output (O_org) “at t1” on KV_0, KV_1, and KV_3 can be merged with it. However, “at t0”, we do not know KV_2 is not going to be selected “at t1”. Hence, we need to compute O_sup using KV_2 “at t1” (not at t0).
>
> As the reviewer stated, we assume that the KV cache is present in the GPU’s off-chip DRAM. However, “at t1”, KV_0, KV_1, and KV_3 are fetched to the GPU’s on-chip memory (like L2 cache), while KV_2 is not. To compute the O_sup using KV_2 “at t1”, we need to re-fetch KV_2 from the GPU DRAM to its on-chip memory, which proportionally increases the memory-loading time and is indeed the same as increasing the actual KV cache budget. As our goal is to improve the performance without increasing the actual KV budget (as mentioned in L53), we only consider “retrospective” (i.e., supplementation from the current token to previous ones) updates.
>
> ---
>
> **[Q3, Q4 - RetroAttention in distributed decoding / MoE models]. "Whether RetroAttention will work on distributed decoding? How would RetroAttention work on MoE models?"**
>
> **Answer.** We expect RetroAttention would be seamlessly applicable to distributed decoding or MoE models.
>
> We consider different types of distributed decoding, such as tensor parallelism (TP), pipeline parallelism (PP), and data parallelism (DP). As DP simply copies the model into multiple GPU devices, each GPU device will be able to operate retrospective updates as it does in a single GPU. Given that our retrospective update is an attention head-wise operation, it is not influenced by either TP or PP as well, as these methods ultimately shard attention heads or decoding layers.
>
> For Mixture-of-Experts (MoE) models, as MoE is primarily designed with multiple feed-forward networks, we do not expect this will influence or hinder the operation of RetroAttention, which is essentially performed at the multi-head attention layers.

---

### Author Response · Authors · 2025-11-21
**Global Response to the Reviewers**

Thank you for the valuable and constructive feedback. We sincerely thank all reviewers for their thoughtful evaluations and constructive feedback. We are encouraged that reviewers highlighted several key strengths of our work, including: the importance of long-generation refinement and clear motivation (zHKN, A5Uv), effectiveness and consistency of the performance gains (A5Uv, ytzA, zGZu), novelty and clarity of the retrospective update mechanism (zGZu, A5Uv), and system-awareness and explicit overhead analysis (zHKN, ytzA, zGZu).

---

Multiple reviewers raised concern regarding the breadth of evaluation across larger or more diverse models and retrieval strategies. To address this, we expanded our experiments to include Qwen2.5-14B and Qwen2.5-32B (see Table 2), both of which show consistent improvements when integrated with RetroAttention.

Reviewers also asked about potential overhead and serving-level practicality. Our new throughput measurements across various batch sizes and context lengths (Table 6) confirm that throughput reductions remain small (typically 3%), and the corresponding latency overhead is minimal (2ms/token at w=8). Importantly, RetroAttention introduces no additional memory traffic for KV cache, and thus preserves the memory-bound execution regime, ensuring that the method integrates cleanly into realistic long-generation pipelines.

---

We thank the reviewers again for their time and efforts throughout the review process.

---

### Summary of primary updates
(main)
- In Section 2.2, a new paragraph “Long-generation tasks are more sensitive to KV budget” is added.
- Table 2 (Comparison in larger models) is added.
- In Section 3.2, a new paragraph “Accuracy Improvements in Larger Models” is added.
- In Section 3.4, a new paragraph “Performance Gain Saturation” is added.

(Appendix)
- A. Overview: anonymized code link is updated.
- new section “B. Motivational Study” is added.
- Figure 8 (Token distribution analysis in long generation) and Table 4 (Comparison against top-K reusage) are added.
- Table 5 (Relative attention-score gain by retrospective updates) is added.
- new subsection “C.3. Supplemented Attention Weights” is added.
- new paragraph “Batch-size impacts on decoding throughput” is added.
- Table 6 (Batch-size impacts on throughput) is added.
- new subsection “F.3. RetroAttention with other Method” is added.
- Table 9 (Comparison with other method (ArkVale)) is added.
- new subsection “C.4. Sensitivity to Retrospective Window” is added.

---

### Author Response · Authors · 2025-12-03
**Summary of Rebuttal**

Dear Area Chair,

We sincerely appreciate the time and effort that AC and the reviewers dedicated to evaluating our submission. Below is a concise summary of the discussion during the rebuttal period, highlighting the overall reviewer response and the main issues we addressed.

> **Overall Reviewer Response**

- Among the four reviewers, three provided clearly positive assessments, with scores of **8 (ytzA), 6 (zHKN), 6 (A5Uv), and 2 (zGZu)**.

- **Reviewer ytzA** emphasized both the novelty and practical impact of our approach and, after the discussion, expressed optimism about the contribution. **Reviewer A5Uv** highlighted our clear motivation, empirical validation, and novel mechanism for long-context refinement, and **Reviewer zHKN** further noted the potential of RetroAttention to be broadly applied to many backbone sparsification strategies.

- **Reviewer zGZu** raised concerns primarily regarding empirical motivation. In our rebuttal and revision, we directly engaged with these concerns and added further analyses and experiments to validate our modeling choices.

- **Reviewer zHKN and A5Uv** requested additional ablation studies, evaluations on larger models, and experiments combining our method with other KV cache compression methods. We addressed all of these during the rebuttal, and the revised manuscript now includes the requested experiments.

> **Major Points Addressed in the Rebuttal**

- We demonstrated that **RetroAttention improves the accuracy in larger and architecturally different models**, including Qwen2.5-14B and 32B. These results consistently reinforce our main claims and are included in the revised manuscript (Table 2; L406-L412).

- We addressed Reviewer zGZu’s concerns about empirical motivation for RetroAttention. We quantify how often adjacent queries miss non-overlapping top-k pages and highlight that **long-generation tasks are substantially more sensitive to KV budget than long-input tasks**. This supports that retrospective KV updates within a sliding window can yield non-marginal improvements for long-generation performance.

- To address concerns regarding latency and system complexity, we added decoding-throughput benchmarks across multiple batch sizes and long-context settings (Table 6; Appendix D). The results show that **RetroAttention maintains high throughput under practical batch sizes**.


We remain grateful for the constructive feedback provided by all reviewers and for your oversight throughout this process.

Best regards,

Authors

---

### Meta-Review · Area_Chair_jTdj · 2026-01-03

**Summary:**

In conventional autoregressive LLM setting, once a token is generated, the model never revisits it or fixes how it understood it in the past. If something important appears later, earlier tokens can’t benefit from it. The core proposal of this work is the following: RetroAttention lets the model go back and update how recent past tokens attended to the context, using new information that appeared later. The work demonstrates strong results showing that the effective KV exposure can be larger than the cache size.

3 out of 4 reviewers expressed support for this work. The main concern, stated by the remaining reviewer (zGZu), pertains to limited potential benefits of the proposed method given that the paper adds a complicated “go back and update recent tokens” mechanism, but in practice those recent tokens may already be looking at the same information, since nearby tokens are usually strongly correlated. Thus, the proposed method might not actually add significant benefits.

The authors disagreed with reviewer zGZu during the rebuttal, and argued that their method demonstrates benefits empirically. During the allocated discussion time the authors and the reviewer did not reach agreement.

Judging the submission holistically, and anticipating the course of the discussion if additional time was available, I believe the submission is innovative enough and proposes a new method, which has not been considered before. While evaluating the breadth of applicability of this method may require additional work, the results and ideas as they already appear in the paper would benefit the ICLR community. For this reason, I recommend accepting this work.

This recommendation is contingent on inclusion of a comprehensive discussion of the relationship between Yang et al., 2025 (LServe: Efficient Long-sequence LLM Serving with Unified Sparse Attention) and the present paper in the main text of the camera-ready version, as requested by reviewer zGZu. Currently, this discussion appears only in Appendix B, but not the main text.

**Reviewer Concerns:**

3 out of 4 reviewers expressed support for this work. The main concern, stated by the remaining reviewer (zGZu), pertains to limited potential benefits of the proposed method given that the paper adds a complicated “go back and update recent tokens” mechanism, but in practice those recent tokens may already be looking at the same information, since nearby tokens are usually strongly correlated. Thus, the proposed method might not actually add significant benefits.

The authors disagreed with reviewer zGZu during the rebuttal, and argued that their method demonstrates benefits empirically. During the allocated discussion time the authors and the reviewer did not reach agreement.

Appendix B, added during the discussion, addresses most of these concerns.

**Reviewer Scores:**

The submission received support from 3 out of 4 reviewers with scores (6,6,8). The remaining reviewer's (zGZu) score is 2. I believe this score would have improved, if additional discussion time was given.

---

### Decision · Program_Chairs · 2026-01-26

Accept (Poster)